# Flood Risk Assessment Using GIS-Based Analytical Hierarchy Process in the Municipality of Odiongan, Romblon, Philippines

Jerome G. Gacu [1,2,3], Cris Edward F. Monjardin [1,2,4,*], Delia B. Senoro [1,2,4] and Fibor J. Tan [1,2,4]

1    Masters Program in Civil Engineering, School of Graduate Studies, Mapua University,
     Manila 1002, Philippines
2    School of Civil, Environmental and Geological Engineering, Mapua University, Manila 1002, Philippines
3    Civil Engineering Department, College of Engineering and Technology, Romblon State University, Liwanag,
     Odiongan, Romblon 5505, Philippines
4    Resiliency and Sustainable Development Center, Yuchengco Innovation Center, Mapua University,
     Manila 1002, Philippines
*    Correspondence: cefmonjardin@mapua.edu.ph

**Abstract:** The archipelagic Romblon province frequently experiences typhoons and heavy rains that causes extreme flooding, this produces particular concern about the severity of damage in the Municipality of Odiongan. Hence, this study aimed to assess the spatial flood risk of Odiongan using the analytical hierarchy process (AHP), considering disaster risk factors with data collected from various government agencies. The study employed the geographic information system (GIS) to illustrate the spatial distribution of flooding in the municipality. Sendai Framework was the basis of risk analysis in this study. The hazard parameters considered were average annual rainfall, elevation, slope, soil type, and flood depth. Population density, land use, and household number were considered parameters for the exposure assessment. Vulnerability assessments considered gender ratio, mean age, average income, number of persons with disabilities, educational attainment, water usage, emergency preparedness, type of structures, and distance to evacuation area as physical, social, and economic factors. Each parameter was compared to one another by pairwise comparison to identify the weights based on experts' judgment. These weights were then integrated into the flood risk assessment computation. The results led to a flood risk map which recorded nine barangays (small local government units) at high risk of flooding, notably the Poblacion Area. The results of this study will guide local government units in developing prompt flood management programs, appropriate mitigation measures, preparedness, and response and recovery strategies to reduce flood risk and vulnerability to the population of Odiongan.

**Keywords:** AHP; digital elevation model; flood; GIS; risk assessment

## 1. Introduction

Floods are caused by the failure of natural paths and drainage systems to hold excess water during and immediately following excessive rainfall [1]. This condition is among the disastrous natural hazards that can cause tremendous economic loss, damage to infrastructures and natural ecosystems, as well as death. The Organization for Economic Cooperation and Development (OECD) reported that floods trigger more than USD 40 billion in destruction worldwide [2]. The United States loses about USD 8 billion a year due to flooding. Recently, casualties have risen to roughly 100 deaths annually [3] and about 6.8 million were adversely affected by excessive flooding in the northeastern part of India. Nepal, Indonesia, and Japan [4].

According to a study by Monjardin et al. [5], flooding is a dangerous natural phenomenon that has taken numerous lives and caused enormous economic damage in the Philippines. Flood is considered the second most frequent calamity in the Philippines, representing 31.9% of annual natural disasters [6]. The National Council for Disaster Risk

Reduction and Management (NDRRMC) of the Philippines reported on 19 April 2021 that 68,490 individuals were evacuated in Bicol and Eastern Visayas regions due to risk from Tropical Cyclone Surigae (Bising) [7]. Additionally, the Mindanao Island that was formerly considered as a region free from typhoons was devastated by consecutive typhoons, e.g., Sendong (international name, Washi) and Pablo (international name, Bopha). These typhoons altered the usual typhoon pathway and made a new typhoon route. These two typhoons landed in 2011 and 2012, respectively, and caused devastation that killed more than 1000 individuals and 100 people went missing [8]. Further, as mentioned in the study of Siddayao et al. [9], Typhoon Haiyan distressed the Philippines telecommunication signals, power, and water lines on 8 November 2013. In the province of Romblon, floods frequently occur, resulting in losses to the affected municipalities. All rivers and tributaries in the Romblon province overflowed [10] during typhoons. Flood occurrences are frequent in the Municipality of Odiongan, being a low-lying area of Tablas Island, Romblon province.

In the Philippines, flood risk maps are essential for the safety of communities and ecosystems [11]. Decision-makers are looking for longer-term mitigation of the adverse effects of floods and some natural tragedies; hence, confidence criteria in engineered solutions such as flood protection systems are important [8]. Furthermore, the assessment and evaluation of flood hazards must be constructed on accurate flood hazard guides to show the real impact of urban development [12]. Risk assessment is vital in formulating decisions guidelines, policies, and mitigations based on meteorological, hydrological, and socioeconomic factors [13]. Comprehensive flood risk assessment and the enhancement of efficient flood mitigation actions need systematic information regarding flood occurrences at points in a catchment basin [14]. However, specific factors of population, society, economy, environment, transportation, and other disaster-bearing elements in different parts of mountain cities are remarkably varied, which increases the doubt of risk assessment index weight and risk assessment reliability [15]. Hence, accurate hazard maps and least-error indices are important tools in risk assessment.

The GIS tool plays a vital component of flood risk assessment due to the evaluation process that needs spatial information. The practice of a standard approach for evaluation and merging distinctive data affect the precision and comparability of assessment outcomes. Some nations have established national guidelines to assess flood risk potential [16]. In addition, GIS can be utilized to study international, regional, and local flood risks and guide the implementation of a risk mitigation plan [17]. GIS in the Philippines is a primary distinctive tool used in countrywide flood risk modeling. However, existing high-resolution flood risk models have come to be very important. These tools can be used for flood readiness by improving these maps' data levels [18]. ArcGIS, developed by ESRI, is a GIS-based tool that can produce standard Web Services and make numerous network GIS uses [19].

The Digital Elevation Model (DEM) is widely used in GIS modeling, and the enhancement, development, and processing of DEMs are vital in many environmental aspects. It is in the form of a grid as a digital illustration of land with a corresponding pixel value equal to an elevation from the datum [20,21]. According to Suguruman et al. [22], DEMs are used more often in flood risk management, including flood plain models, visualization, flood hazard assessment, and identification of floodplain altitudes. There are numerous sources of DEM information, including Advanced Space Borne Thermal Emission and Reflection Radiometer (ASTER), Synthetic Aperture Radar (SAR), Global Positioning System (GPS), and Light Detection and Ranging (LiDAR) [23]. In the Philippines, hydraulic and hydrologic tools for flood risk analysis are very limited in line with topographic, geometric, and hydrologic river information [24]. The Philippines assimilated geospatial data LiDAR and IfSAR (Interferometric Synthetic Aperture Radar) with excellent resolution Digital Terrain Models (DTMs) covering 300,000 square kilometers of the terrestrial area [25]. This is to deal the insufficient high-resolution topographic maps.

Flooding needs considerable attention, studies have evaluated the connection between urban/rural services, flood history, and disaster readiness in local communities living in

safety [26]. The Sendai Framework acknowledged the critical role played by the community in disaster risk reduction [27]. This framework is used in disaster risk management delivers quantifiable parameters for a national and local scale to calculate the reduction in disaster damages. The compilation and evaluation of disaster damages under the Sendai Framework enhance our knowledge of the efficiency of disaster risk reduction approaches [28].

There is a need to understand the spatial extent of flood zones by utilizing multiple data to show a possible baseline for consistent flood risk management and mitigation measures [29]. The methodology using multicriteria analysis (MCA), also known as multi-criteria decision-making (MCDM), supports a basis that can hold distinctive assessment on determining the factors of a composite decision, arrange the aspects into a hierarchical configuration, and analyze the relations amid elements of the identified hazard [30]. All MCA methods make the options and their influence on the different criteria clear. They vary, however, in how they associate all the data needed. The method's primary role is to solve the difficulties that decision-makers have encountered when handling a large quantity of complex information. MCA can be used to recognize a single most preferred option, rank options, shortlist a limited number of options for subsequent detailed evaluation, or differentiate conventional from unconventional possibilities [31].

Several approaches have been suggested for MCA, but the Analytical Hierarchy Process (AHP) is being used most frequently to resolve different flood risk assessments [32]. The AHP provides the same advantage as MCA models in focusing decision-maker consideration on developing a structure to gather all the significant factors expected to differentiate the best option [31,33]. AHP represents the problem in three parts where the first part is the matter that needs to be fixed, and the second part is the alternative solutions available to resolve the problem. The third and most important process is the criteria expended to assess the alternative solutions [34]. Studies on flood risk assessment in Thailand [26,35], Bangladesh [36], and Indonesia [37] used GIS and AHP. Additionally, in the Philippines, identified relevant flood factors and judgments of decision-makers were analyzed using AHP judgments to weigh each parameter in estimating flood hazards in the study [38] at the central business district of Tuguegarao City, Philippines. Another study was conducted in Infanta, Quezon Province, Philippines, aiming to give the municipality options and models for flood mitigation. The drainage system in said municipality is at risk of causing flood-related problems deliberating identified relative factors via AHP [6]. The evaluation of flood zones and flood problems for Davao Oriental, Philippines, were analyzed by the AHP and Maxent tool which reduce the subjectivity and uncertainty in selecting and weighting criteria [8]. The rareness of using AHP-based research made it easier to make a model of indecision without compromising the subjective and objective aspects of the assessment process [29]. Hence, the number of flood events in the Municipality of Odiongan that caused property damage to the community explicitly need the output of this research study. The results of this assessment will be used as the basis for the municipality's flood mitigation and risk management. Additionally, the information will useful in areas with similar topography and weather conditions.

## 2. Materials and Methods

### 2.1. Study Area

The study area is the Municipality of Odiongan located in the middle west portion of Tablas Island, Romblon province with coordinated of 22°04′ East Longitude and 12°19′ North Latitude. Odiongan has a land area of 185.67 square kilometers representing 12.11% of Romblon province. The town proper lies in the low-lying plains, and the interior part of the municipality is composed of hills and mountainous forests. Odiongan consists of 25 barangays and 1 anchorage, which is linked to other neighboring islands. Figure 1 shows the imagery map of Odiongan with barangay boundaries.

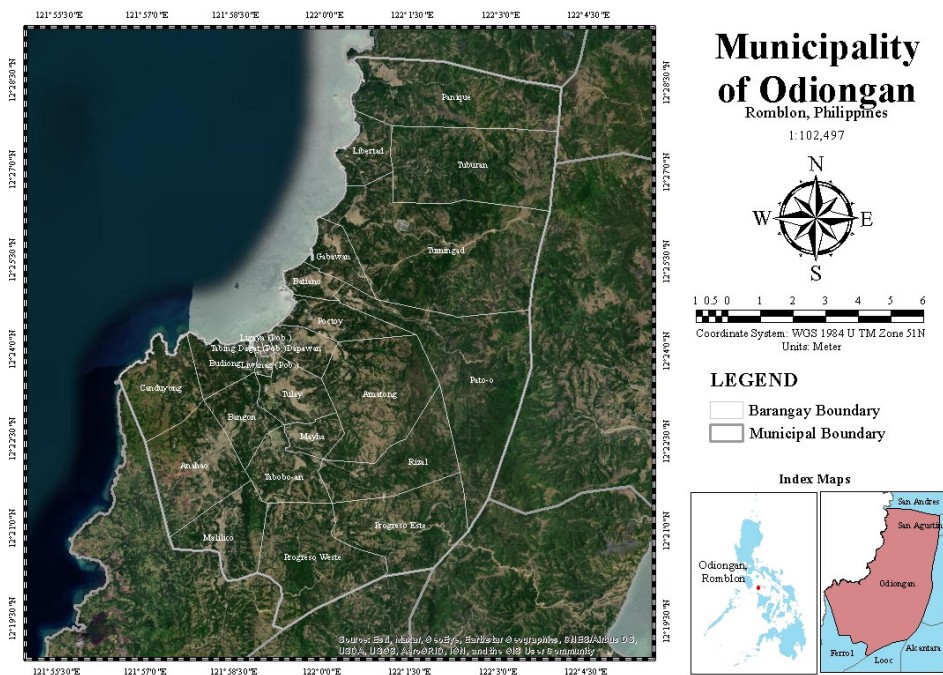

**Figure 1.** Imagery map with barangay and municipal boundaries of the study area.

### 2.2. Data Collection and Identification of Factors

The Sendai Framework was followed to identify flood indicators in assessing flood disaster risk, hence identifying the parameters and the data that need to be collected. There were three identified categories of flood risks parameters, such as (a) hazard, (b) vulnerability, and (c) exposure. The hazard parameters considered were: Average annual rainfall, slope, elevation, soil type, and flood depth. The parameters for vulnerability that were considered were: Gender ratio, age, average income, physical health of the individual, educational attainment, water usage, emergency preparedness, types structures, and proximity to the evacuation center. The parameters for flood exposure were: Population density, number of households, and land use/details are shown in Table 1.

**Table 1.** Parameters with the type of data used, duration/year, and source used for hazard, vulnerability, and exposure assessment.

| References | Parameter | Data Type | Duration/Year | Source |
|---|---|---|---|---|
| **Flood Hazard Parameters** | | | | |
| [8,17,35,39–41] | Average Annual Rainfall | Interpolated Climatological Normal using Isohyetal Method | 2020 | PAGASA and web search for weather station coordinates |
| [8,17,26,32,35,41] | Slope | Derived from IfSAR Data using Slope Tool in ArcMap | 2013 | (NAMRIA-DENR) |
| [6,8,26,32,35,41] | Elevation | Derived from IfSAR Data using Field Contour Tool in ArcMap | 2013 | (NAMRIA-DENR) |
| [8,9,17,42] | Soil Type | Shapefile from the archive of CLUP | 2011 | Municipality of Odiongan, Romblon—(CLUP) |
| [37,43–45] | Flood Depth | 100-year period of flood model simulated in HEC-HMS and HEC-RAS and MGB Flood Susceptibility Map | 2018 | MGB |
| **Flood Vulnerability Parameters** | | | | |
| [39,43] | Gender Ratio | Men to women gender ratio | 2020 | Barangay Profile |
| [43] | Average Age | Mean age of the individual | 2020 | Barangay Profile |
| [31,43] | Average Income | Annual average income per household | 2020 | Barangay Profile |
| [39,46] | Number of Persons with Disabilities | Number of PWD in barangay | 2022 | Barangay Management System (BMS) |
| [43,46] | Highest Educational Attainment | Average educational attainment of individuals in barangay | 2020 | Barangay Profile |

**Table 1.** *Cont.*

| References | Parameter | Data Type | Duration/Year | Source |
|---|---|---|---|---|
| [46–48] | Water Usage | Primary source of water | 2022 | Survey Questionnaire |
| | Emergency Preparedness | Emergency preparedness during unexpected situations like natural disasters | 2022 | Survey Questionnaire |
| [29] | Types of Built-up Structures | Classification of structures of every household | 2020 | Barangay Profile |
| | Distance from the nearest Evacuation Area | Distance of identified evacuation area using buffer tool in Arcmap | 2022 | Site Investigation and Survey Questionnaire |
| **Flood Exposure Parameters** | | | | |
| [9,29,40,42,49] | Population Density | Computed from the population over the covered area of the barangay | 2020 | PSA |
| [43] | Household Number | Number of households of every barangay | 2020 | PSA |
| [6,17,26,35,40,41] | Land Use/Land Cover | Land cover map from CLUP | 2011 | Municipality of Odiongan, Romblon—(CLUP) |

### 2.2.1. Flood Hazard Parameters

Flood management cannot be adequately completed without assessing flood hazards [48]; therefore, details of indicators are elaborated below.

1.  Average Annual Rainfall

Precipitation values were plotted on a suitable base map at their respective stations using isohyetal method, and isohyets were drawn to create an isohyetal map. The study used the climatological normal records [50] from long-term averages over 30 years of PAGASA weather stations (Figure 2) with corresponding coordinates. Spatial interpolation employing the isohyetal method was applied to obtain dimensional rainfall patterns for projections of Romblon.

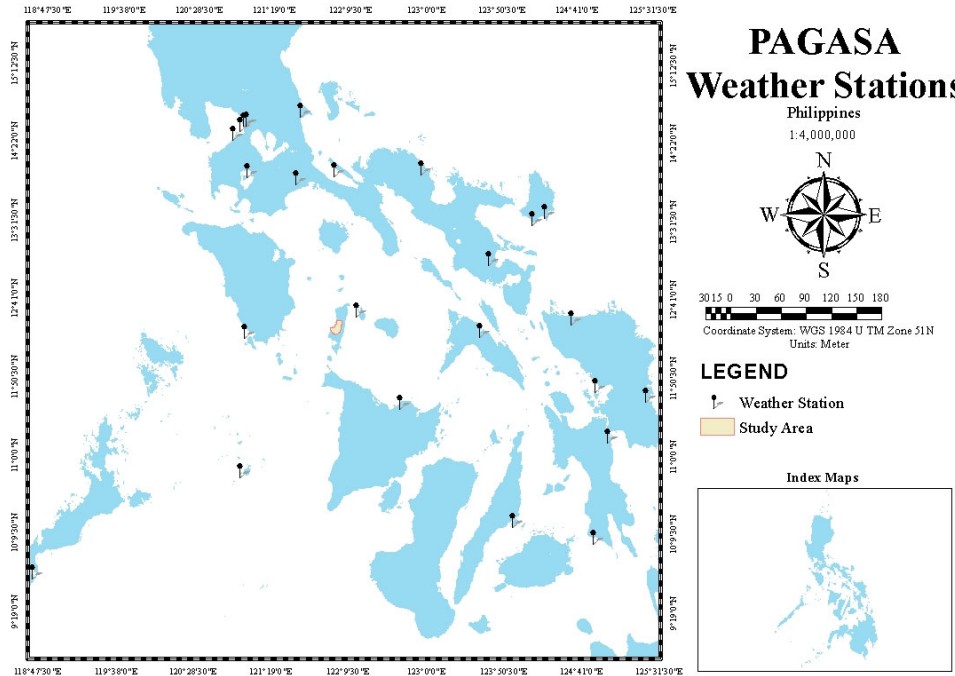

**Figure 2.** Weather stations considered in the interpolation for average annual rainfall using the Isohyetal Method.

2.  Slope

The slope is a critical factor contributing to the intensity of destructive forces of floods in a particular area. The study prepared the slope map using the IfSAR DTM from National Mapping and Resource Information Authority (NAMRIA) and the spatial tool in the GIS application platform.

3. Elevation

Ground elevation is one of the main factors that should be considered in assessing flood hazards. IfSAR data were utilized and processed in the GIS tool.

4. Soil Type

The study used the soil map based on the map of NAMRIA stipulated in the Comprehensive Land Use Plan (CLUP) [51] of the Municipality of Odiongan. These data were correlated to the soil's water holding capacity and infiltration rate.

5. Flood Depth

ArcGIS, HEC-HMS (Hydrologic Engineering Center's—Hydrologic Modeling System) [52] and HEC-RAS (Hydrologic Engineering Center's—River Analysis System) [53] were the tools used for flood hazard simulation. The most important data used in the simulation were the DEM, which were provided by NAMRIA with a resolution of five-by-five ($5 \times 5$) meters. A combination of simulated maps and Flood Susceptibility Maps [54] from the Mines and Geoscience Bureau (MGB) were used in the study.

### 2.2.2. Flood Vulnerability Parameters

The vulnerability factor includes social, economic, and personal safety [44]. Demographics and disaster risk reduction data of the Municipality of Odiongan were gathered through actual surveys (questionnaire) and existing records of the local government of Odiongan.

1. Demographics

The demographic data were gathered from the database of Philippine Statistics Authority (PSA). The period considered was 2015 to 2020. The on-site survey was conducted in every barangay of the Municipality of Odiongan.

2. Disaster Risk Reduction Data

All data were extracted from the survey conducted in every barangay and CLUP of the Municipality of Odiongan. The barangay identified evacuation facilities where coordinates and floor areas were recorded using GPS and area measuring tools.

### 2.2.3. Flood Exposure Parameters

The exposure analysis was aimed at identifying the life and property elements exposed in flooding events [41]. The identified exposure elements were population density, number of households, and land use/cover. The data were taken from the PSA record and municipal zoning maps archived from CLUP of the Municipality of Odiongan.

### 2.3. Modeling, GIS Mapping, and Validation

Generated models and maps from ArcGIS were the primary basis in the computation and analysis of final flood risk indices.

### 2.3.1. Basin Model Pre-Processing

In creating a basin model of Odiongan River channels (Bangon River), IfSAR-DEM with a 5 m $\times$ 5 m resolution was used. Data were processed using the GeoHMS10.7 tool plugin in ArcGIS 10.7. This is to generate a basin model and incorporated with the available soil and land cover data of 2004 from NAMRIA to assign curve numbers (CN) for each sub-basin. Soil type and land cover classification were represented as CN for each sub-basin. Initial abstraction (IA), time of concentration (TC), Storage Coefficient (SC), River Length, and sub-watershed area were derived during the pre-processing of the basin model.

### 2.3.2. Basin Model Calibration and RIDF Simulations

The pre-processing output of the HMS Basin Model was calibrated under the HEC-HMS 4.9 software to model the hydrologic response of the watershed to a specified hy-

drometeorological input. The parameters' values were tuned to attain an at least acceptable result in all the statistical measures recommended for model evaluation. As the model was calibrated, the simulations of rainfall scenarios of 25-, 50-, and 100-year followed. The rainfall intensity duration frequency (RIDF) data were acquired from PAGASA Romblon, Romblon province rain gauge station with 48-year rainfall records. These data were entered as the meteorological model file using the frequency storm precipitation method in HEC-HMS performed with calibrated basin model. The outputs of the simulations were then calibrated basin model with precipitation and outflow data of the three (3) return periods.

### 2.3.3. Two-Dimensional (2D) RAS Model Simulations

The processed DEM was used to create the river analysis model (RAS) model using the HEC-RAS 6.2, a practical river hydraulic simulation and analysis software. The RAS model was processed through unsteady flow analysis, and the boundary conditions used were flow hydrograph in the upstream and normal depth in the downstream which considers both the frictional resistance and slope of the channel. The calibrated outflow in HMS and precipitation were incorporated into the model. Flood depth considering a 100-year return period was regarded as one of the parameters in hazard mapping; this was exported as raster files and translated into spatial data in the GIS.

### 2.4. Evaluation and Assessment of Parameters Using AHP

Contributing factors were identified and assessed in which the weights of each parameter were determined using AHP based on the knowledge of experts composed of end-users, hydrologists, meteorologists, water resource engineers, and persons with comprehensive expertise in disaster risk reduction. Experts from government agencies such as PAGASA, Bureau of Soils and Water Management (BSWM), and MGB participated in the survey. A specialist from academic institutions (University of the Philippines, Mapúa University, Central Luzon State University, and Asian Institute of Technology) and an end-user (LGU-Odiongan) also responded to the survey. The survey for pairwise comparison was delivered and requested thru an online and printed-out questionnaire. A risk assessment was proceeded using the weights of each factor derived in AHP through a pairwise comparison questionnaire.

### 2.4.1. Determination of the Priorities among the Decision Elements of the Hierarchy

The feature weights were assigned parameters, where levels were reclassified and normalized into 1 for the least priority and 5 for the most focused. This step gathered the weight for each criterion and option using a pairwise comparison technique. Ten (10) experts on-field and end-users participated in determining the relevance of one alternative over the other with a pairwise comparison method presented in a matrix.

Each comparison was graded by experts and end-users using the pairwise comparison technique scale. The procedure usually contains a questionnaire for comparing all the elements and a geometric mean to arrive at a final solution [32] specifying the nine points intensity matrix, as shown in Table A1 of Appendix A.

### 2.4.2. Derivation of the Overall Relative Weights

The relative significance or weight of the factor after a pairwise comparison matrix was computed based on systematic AHP assessment and expert's inputs. This step was conducted by calculating the normalized values for each criterion and alternative, and choosing the normalized main priority vectors. Normalized values for each criterion and alternative in their respective matrices were derived by dividing each cell into its column and producing a total column of 1 for each criterion and alternative. Weights were calculated by averaging the rows of the matrix. The resulting value will give relative weight to every criterion concerning the best goal, and provide relative weight for the alternatives with respect to the criteria. The final relative weights of the alternatives were defined by computing the product's linear combination (LC) between the relative weight

of each criterion and the alternative for the specific criterion. The decision-makers choose the best according to the alternatives' overall weights if the experts' judgments are proven consistent. This is mathematically expressed using Equation (1).

$$C = \{C_j | j = 1, 2, \ldots, n\} \tag{1}$$

The pairwise comparison on the criteria can be generalized using an evaluation matrix $A$, as shown as Equation (2), in which every element is the quotient of weights of the criteria given in Equation (3) [32].

$$A = \begin{bmatrix} a_{11} & a_{12} & . & a_{1n} \\ a_{21} & a_{22} & . & a_{2n} \\ . & . & . & . \\ a_{n1} & a_{n2} & . & a_{nn} \end{bmatrix}, \ a_{ii} = 1, a_{ji} = \frac{1}{a_{ji}}, \ a_{ij} \neq 0 \tag{2}$$

### 2.4.3. Verification of the Consistency of Judgments and Conclusions according to Results

AHP's quality output was related to the consistency of the pairwise comparison judgments. This step was essential to identify the consistency of the assessment by computing the consistency ratio (CR) before a decision was completed. However, if the problem was expected during deliberation for choosing the best alternative, the CRs for matrices were computed initially before the alternatives' overall relative weights were calculated. After which, calculations were performed to obtain the largest eigenvalue, consistency index (CI), CR, and normalized values for each criterion and alternative.

The last mathematical process normalized and identified the relative weights per matrix. The right eigenvector gave the relative weights ($w$) conforming to the highest eigenvalue ($\lambda_{max}$), as shown in Equation (3).

$$A_w = \lambda_{max} \tag{3}$$

If the pairwise comparisons were consistent, the matrix $A$ was ranked one and $\lambda_{max} = n$, so the weights can be taken by normalizing any of the rows or columns of $A$ [32]. The relativeness between the entries determines the consistency, and the $CI$ was calculated using the equation below:

$$CI = (\lambda_{max} - n)/(n - 1) \tag{4}$$

The final $CR$, which enables the decision-maker to accomplish whether the assessments were adequately coherent, was computed as the $CI$'s and the random index ($RI$) quotient using Equation (5).

$$CR = CI/RI \tag{5}$$

One recommendation for this step was: if the proportion exceeds 0.1, the judgment was considered inconsistent. Therefore, a consistency ratio must be below 0.1 or 10%. The process was reiterated if the evaluation was unpredictable until the CR was within the wanted scale. The user formulated a conclusion according to the assessment results [32].

### 2.5. Development of Flood Risk Map

In this study, Sendai Framework was the basis to evaluate the flood risk by integrating the three (3) criteria, e.g., hazard, vulnerability, and exposure. The result of the flood risk assessment was laid into a map for a better comprehension of it. The last phase of the methodology was to overlay the analysis technique using ArcGIS. The GIS tool generated two or more different thematic maps of a similar area. It overlapped them on top of one another resulting in a new map using the weighted overlay tool. This technique results in a calculation matrix that defined the primary change forms in a study location [26]. The weighted overlay analysis results were developed employing equal intervals with four (4) levels (very low, low, moderate, and high). Flood risk map results were also validated by

doing actual ground assessment of the localities in Odiongan, Romblon, and by reviewing identified flood zones in the area using records of historical flood events.

## 3. Results

### *3.1. Data Analysis for Identified Parameter*

The result of data collection gathered through related literature, and past research studies were put into maps and analyzed. Each map has a scale of 1:100,000 and mainly focuses on identified parameters enumerated in Table 1 and detailed in Section 2.2. Subsequent sections elaborated the results.

#### 3.1.1. Flood Hazard Parameters

Figure 3 shows the maps for every parameter of the Municipality of Odiongan, Romblon province, based on hazard criteria. It was noted that 2203.9 mm was the recorded average annual rainfall by the Romblon Weather Station. The average annual rainfall of the municipality was classified according to interpolation. Figure 3a shows the generated map from ArcMap using the Isohyetal Method with an average annual rainfall ranging from 2200 to 2250 mm. The amount of rain intensifies from the eastern part to the western part of the municipality.

The result of the reclassified slope layer was presented in Figure 3b and categorized in degrees where the green color means the lowest elevation. At the same time, the red part indicates the highest slope. Most maps show a higher slope ranging from 18 degrees to 50 and above. Residential areas were located in the plain areas (green part) where water accumulated during excessive rainfall.

Figure 3c shows the elevation map of the study location extracted using the IfSAR DTM. The elevation was classified into five (5) levels ranging from 0 to 600 m. Most of the map shows a high elevation of 21 to 600 m. The eastern part of the municipality, where the residential and commercial area was located, has the lowest elevation value, varying from 0 to 20 m. These elevations affect how rapidly stormwater could be drained into the catchment based on its slopes.

The slope map shown in Figure 3d was prepared as a shapefile from the Odiongan CLUP 2015. The map was sorted according to its Hydrologic Soil Group (HSG). Soils were classified by the Natural Resource Conservation Service based on the soil's runoff potential. Group A-class (sand, loamy sand, or sandy loam types of soils) has low runoff potential and high infiltration levels, even when fully saturated. They contain chiefly deep, well-drained to excessively drained sands or gravels, and have a high rate of water transmission. Group B is silt loam or loam. It has a moderate penetration rate when fully saturated and consists of moderately deep to deep, well-drained soils with relatively fine to coarse textures. Group C soils are sandy clay loam. They have low infiltration levels when thoroughly wetted. They consist chiefly of soils with a layer that impedes the downward movement of water and soils with moderately fine to fine structure. It is observed that most of the map falls under Group C. The eastern part of the municipality is mainly silt loam or loam. The map shown in Figure 3e is the overlayed flood depth map combined with the MGB Flood Susceptibility Map (see Figure A1) and further discussed below.

In the simulation or modeling process, as a result of delineation, there were 32 watersheds, 16 junctions, and 16 reaches extracted, as shown in Figure 4.

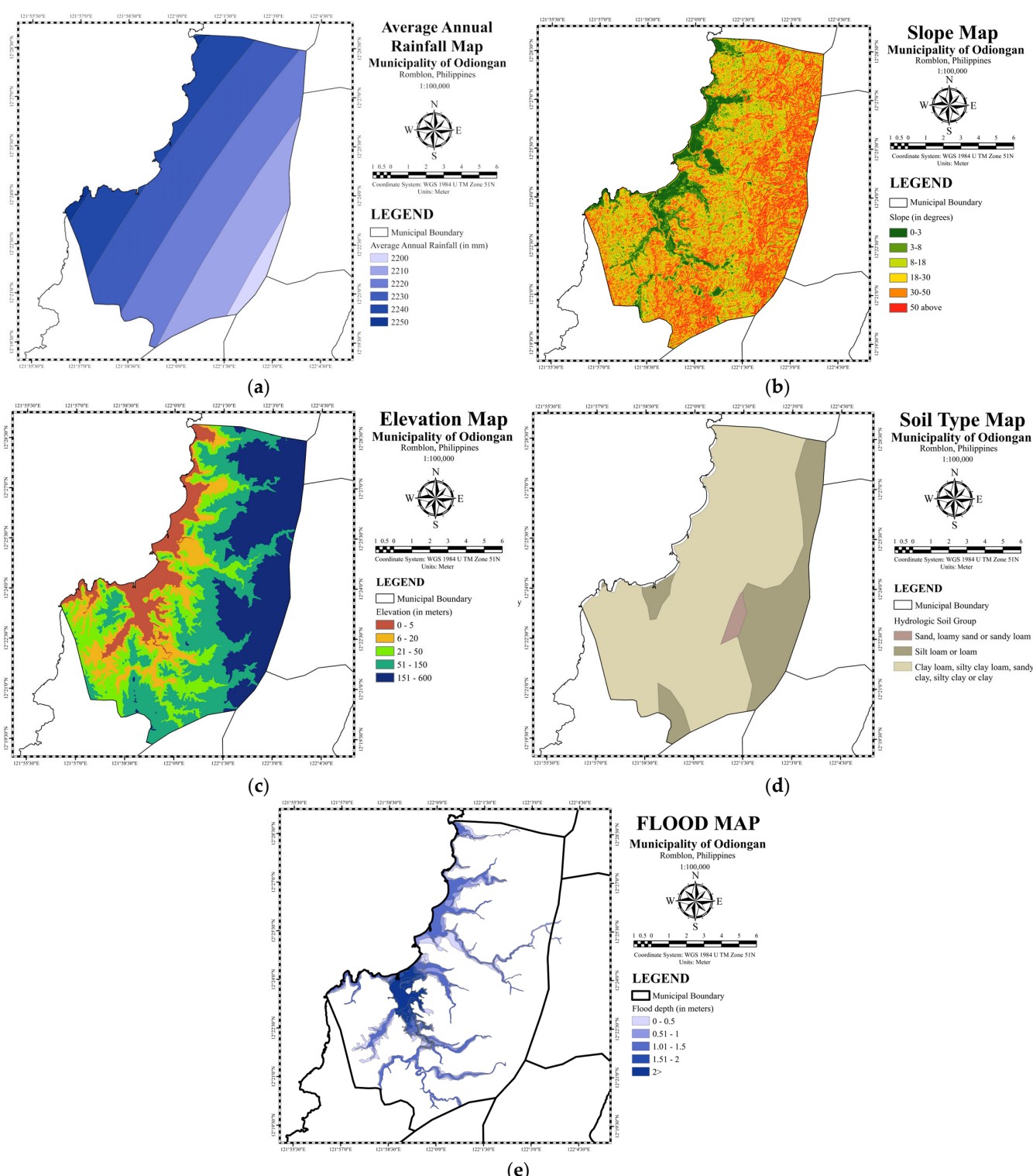

**Figure 3.** Maps generated using ArcGIS in flood hazard parameters: (**a**) Average Annual Rainfall Map; (**b**) Slope Map; (**c**) Elevation Map; (**d**) Soil Type Map; and (**e**) Flood Depth map of Odiongan, Romblon.

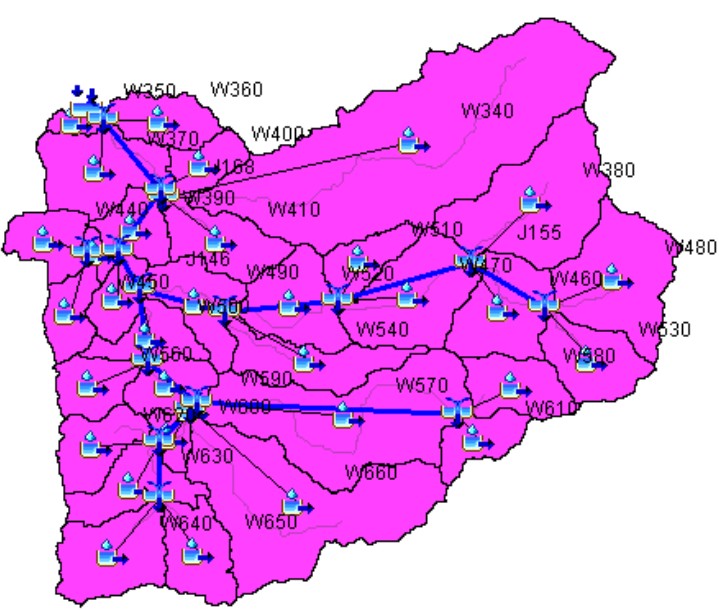

**Figure 4.** Extracted Basin Model of Odiongan, Romblon using GeoHMS10.7 tool in ArcpMAp.

The results from the basin model for 25-, 50-, and 100-year (Figure 5) simulation were exported to excel for the data preparation for hydraulic modeling in HEC-RAS and recorded 2.9 m$^3$/s as its total highest inflow.

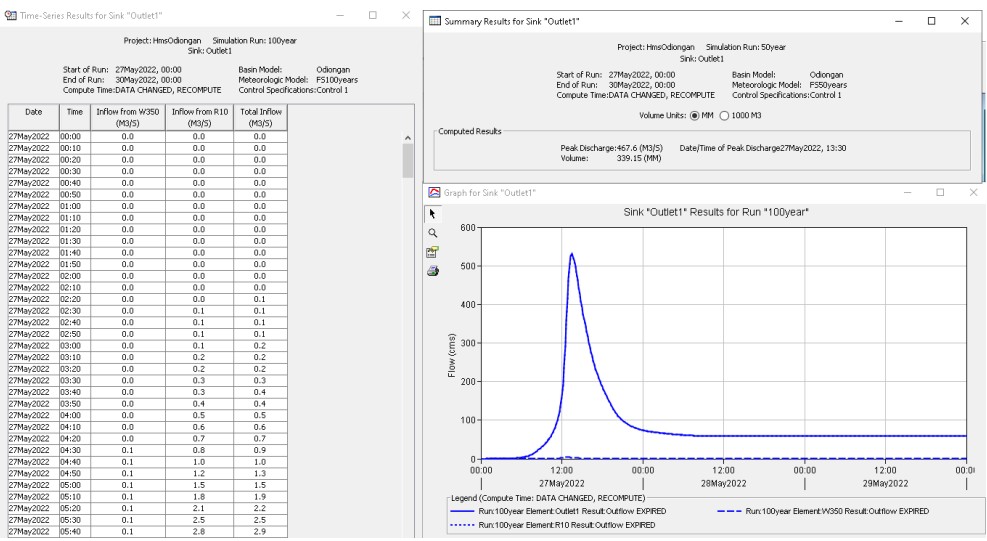

**Figure 5.** Simulation of total inflow results in 100-year using HEC-HMS.

Figure 6 shows that flood depth for all the periods led to a drastic result in a possible flood for the Poblacion area and nearby barangays from the watershed. Based on the simulation, about 5.87 m of flood height were recorded in the worst-case scenario of a 100-year return period. Flood depth with the 100-year model, considered one of the parameters, was exported as raster files from HEC-RAS and mapped to the layers in GIS.

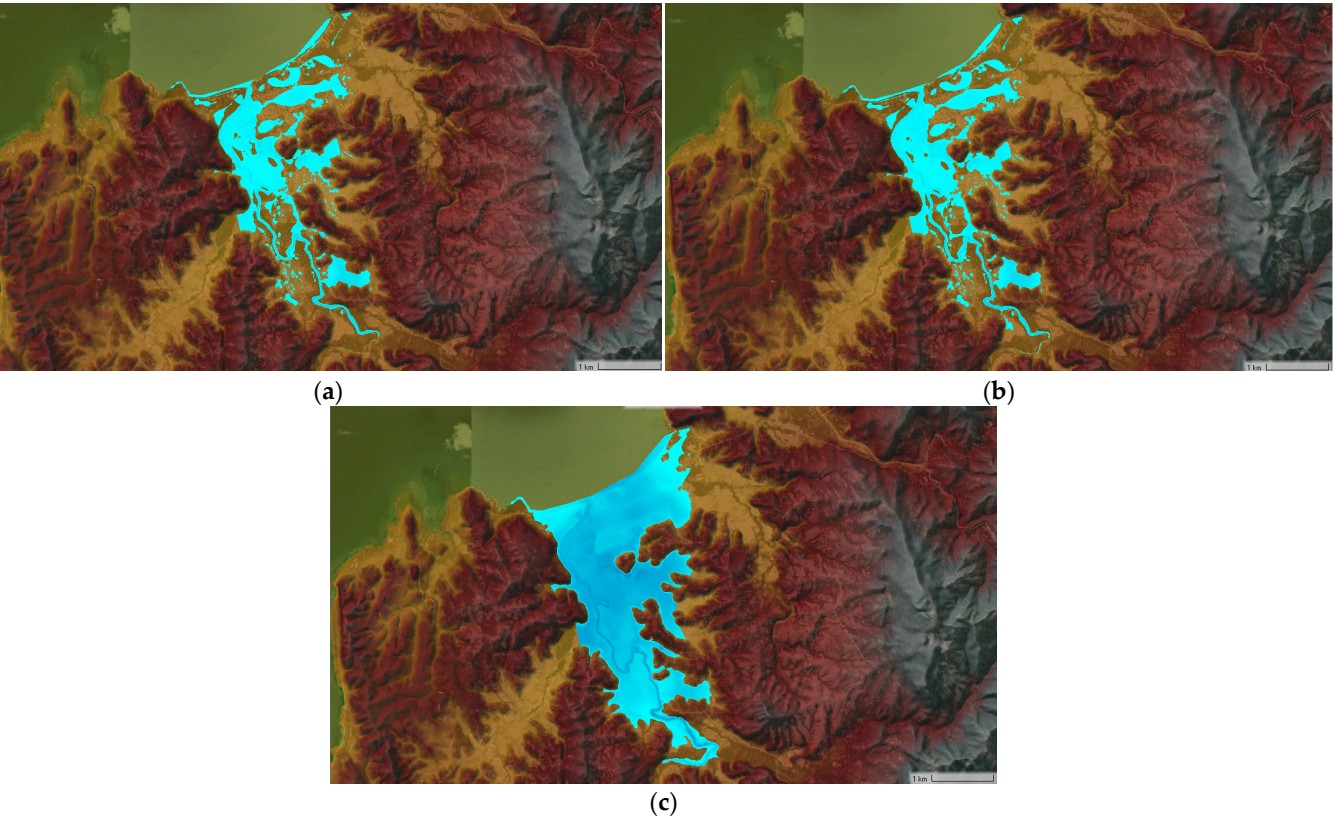

**Figure 6.** Simulation of maximum flood depth in the main river (Bangon River) of Odiongan, Romblon using HEC-RAS for (**a**) 25 years, (**b**) 50 years, and (**c**) 100-year period.

Modeled flood depths were categorized into five (5), e.g., (1) 0–0.5 m (low), (2) 0.5–1 m (moderate), (3) 1.01–1.5 m (high), (4) 1.51–2 m very high, and (5) 2 m and above is considered extremely high. These ranges were based on the Mines and Geosciences Bureau's Flood Susceptibility maps wherein low susceptibility can experience flood heights of less than 0.5 m and a flood duration of less than one (1) day. These included low hills and gentle slopes. It has also spared moderate drainage density. Moderate susceptibility areas were expected to experience flood depths of 0.5 m to 1 m. These spaces are prone to widespread inundation (flooding) throughout long and extensive heavy rainfall and extreme weather conditions. In areas of high susceptibility, where flood height is 1 meter or more with a time of recession of 3 days, are immediately flooded during heavy rains of several hours. The map indicated that most of the area has an adequate slope where only exposure to flood happens in the low-lying zone. Based on the 100-year flood model, the flood surge was concentrated in the town proper of Odiongan with a depth of 3 m for rainfall that occurred in two (2) days based on simulations.

### 3.1.2. Flood Vulnerability Parameters

This study considered the demographics and disaster risk reduction data of the Municipality of Odiongan. Figure 7 shows the maps from the available archival and survey data showing the population age, gender ratio, average income, physical health of the individual, educational attainment, emergency preparedness, and variety of built-up structures.

The men to women gender ratio in the Municipality of Odiongan, as shown in Figure 7a, has recorded more women than men. Barangay Amatong, Rizal and Progresso Este showed only a majority number of men to women. For the total men-to-women gender ratio of Odiongan, it was recorded that the population of women and men is almost the same with a ratio of 0.99991.

The mean age of an individual was considered a parameter. The numbers were taken from each barangay database. Mean age was calculated for all the group data as per age category. As per the results, as shown in Figure 7b, the majority of participants were between 30 and 39. Barangay Pato-o, Amatong, Dapawan, and Bangon have the lowest age range of 29 to 30, while Anahao obtained the highest mean age with 37 to 38 age level.

The average income of individuals was mapped per barangay stipulated in each barangay profile. As shown in Figure 7c, the average annual income per barangay was classified under six (6) levels. Most of the average income ranges from 100,000 to 500,000. However, Barangay Amatong, Bangon, Anahao, Malilico, and Progresso Este have the lowest income, having 40,000 and below.

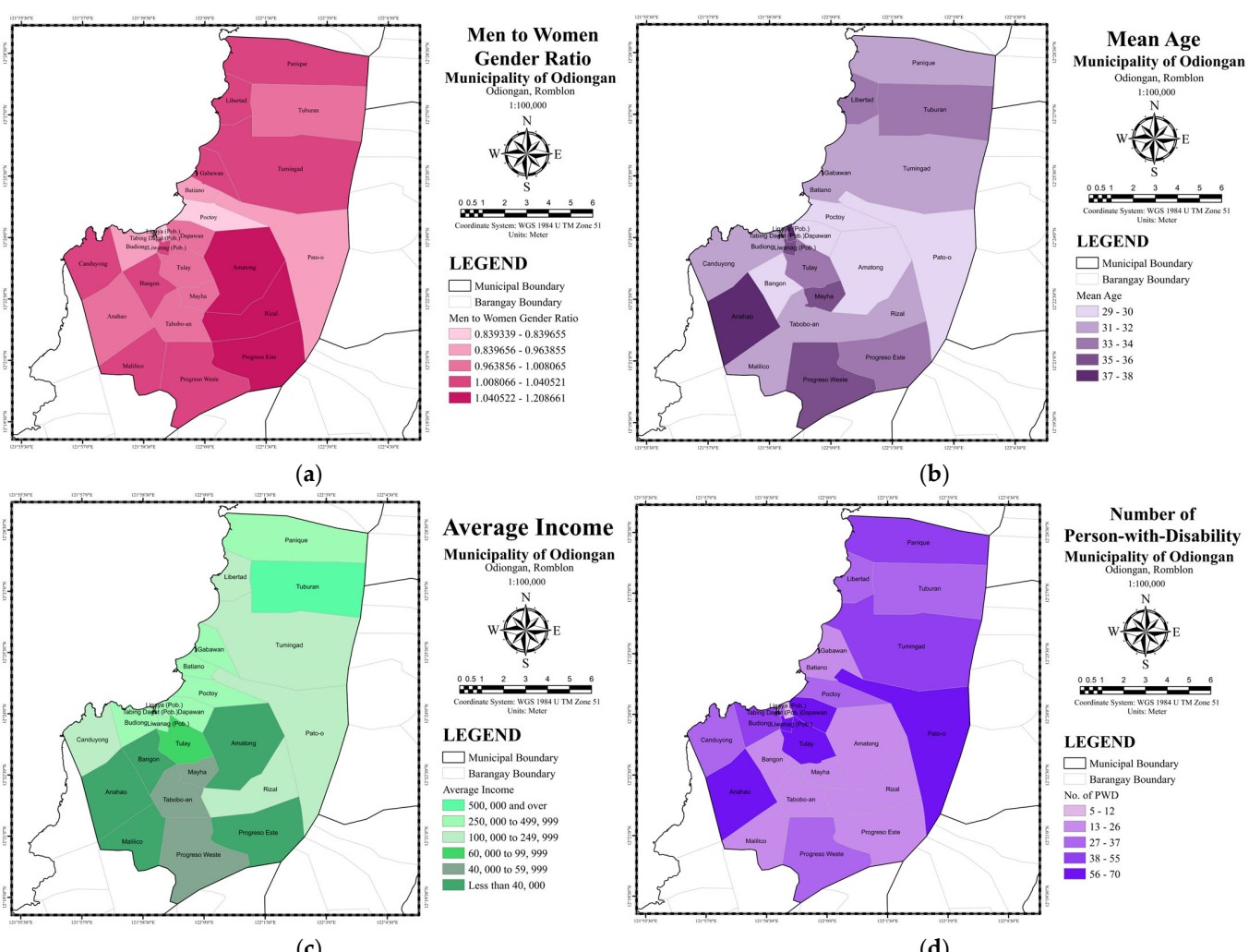

**Figure 7.** *Cont.*

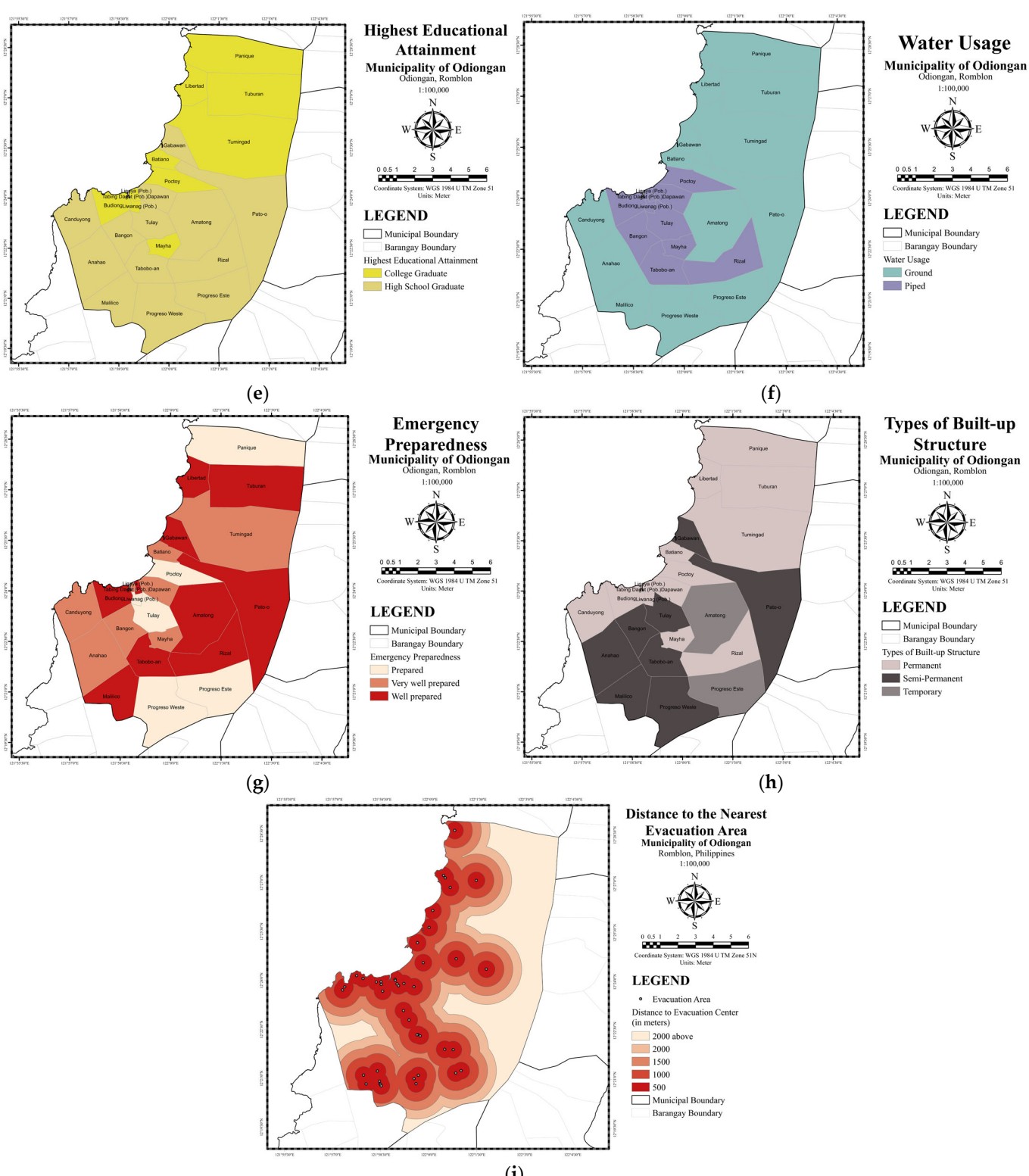

**Figure 7.** Result maps for flood vulnerability parameters in Odiongan, Romblon using ArcMap: (**a**) Gender Ratio; (**b**) Mean Age; (**c**) Average Annual Income; (**d**) Number of PWD; (**e**) Highest Educational Attainment; (**f**) Water Usage; (**g**) Emergency Preparedness; (**h**) Types of Built-up Structure; and (**i**) Distance to the Nearest Evacuation Center.

1.  The floods that hit the areas have disrupted public health and are a subject that has become increasingly important daily due to society's reactions to hazards [55,56].

With a lack of health information, the study considered the number of PWD in each barangay. Under RA 10524, it refers to individuals who agonize long-term physical, mental, intellectual, or sensory impairments that may obstruct their full and practical involvement in society on an equal basis upon interaction with various barriers. The seven types of disabilities mentioned in RA No. 7277 are psychosocial disability, disability due to chronic illness, learning disability, mental disability, visual disability, orthopedic disability, and communication disability [56]. The number of PWD was based on the Barangay Management System (BMS). Records of the number of PWD in Odiongan, Romblon, as shown in Figure 7d, signify a risk in flood events or other natural disasters. The highest number of PWD are in Barangay Anahao, Pato-o, Tulay, and Dapawan, where Tulay and Dapawan experience flood events every year. However, many PWDs were observed in the southern and northern parts of the municipality, excluding barangay Anahao.

2.  The higher the level of education a respondent from a household has, the more likely the individual evacuates [57]. The data used in the study were based on a questionnaire survey conducted in each barangay. Only two categories recorded the highest educational attainment in Odiongan, Romblon, as shown in Figure 7e. More than half of the barangays indicate some high school graduate, and almost half were categorized as college graduates.

The research also incorporated water usage (Figure 7f) as a vulnerability parameter as an additional parameter in disaster risk reduction information. The source of information was a questionnaire survey where the head of barangays was asked for the primary source of water supply. Information was also verified in the records and data of Odiongan Water District. From Barangay Rizal down to the Poblacion (Dawapan, Liwanag, Liwayway, Ligay, Tabin-Dagat) area have access to pipe water. According to the data validated from the Odiongan Water District, 11 out of 25 barangays were supplied by piped water, where the main tank and reservoir are from Barangay Rizal. However, as Romblon is given such a water source, water supply from electric pumps and wells was installed for some barangays.

The emergency prepared data were based on a survey questionnaire's knowledge and input from the head of barangays. The emergency preparedness map is shown in Figure 7g and was classified into three (3) categories, e.g., prepared, well prepared, and very well prepared. It was noticed that only 5 out of 25 barangays, namely, Progresso Weste, Progresso Este, Tulay, Poctoy, and Panique, have been categorized as "prepared" barangays during calamities.

3.  Type of Built-up Structures

The combination of information was taken from the barangay profile as of 2018, and a questionnaire survey was conducted. Figure 7h shows the types of built-up structures and are classified into three (3) categories. These are the (a) permanent, (b) semi-permanent, and (c) temporary shelters. Permanent buildings are structures with concrete foundations and walling, GI sheets as roofing, and other solid materials. Semi-permanent structures are a combination of lumber and concrete elements. Temporary shelters use sawali, bamboo, nipa, and cogon as construction materials. The map shows that most of the barangay have permanent and semi-permanent structures. This indicated that most of the homes in Odiongan are more resilient in terms of flood events. However, eight (8) barangays in the elevated area have residential structures categorized as temporary shelter. Consistent findings were proven in the previous research regarding households' housing types as a significant factor in flood vulnerability [43].

Coordinates were noted and listed during site investigations and measured the floor area to estimate the capacity of every room area during calamity (see Figure A2). Figure 7i shows the ideal coverage of every evacuation area in the municipality on which 500 to 2000 m circles around each evacuation center were drawn and categorized into five (5) levels. The map shows the number of people who can reach the facility within an acceptable

walking distance. In this map, the ideal number of people have been identified for each evacuation center during a disaster.

The Sphere standards imply that in the instant aftershock of a disaster, particularly in dangerous climatic conditions where quarter materials are not readily available, an area of no less than 3.5 square meters per person is suitable to save lives and provide adequate short-term shelter. As an evacuation center is utilized preferably only for a short duration, a center's maximum 'event sheltering' capacity should permit no less than 1.5 square meters per person [55].

### 3.1.3. Flood Exposure Parameters

There were three (3) identified parameters for flood exposure assessment. Figure 8 shows the population density data, land use map, and household numbers from the CLUP Odiongan and PSA Database.

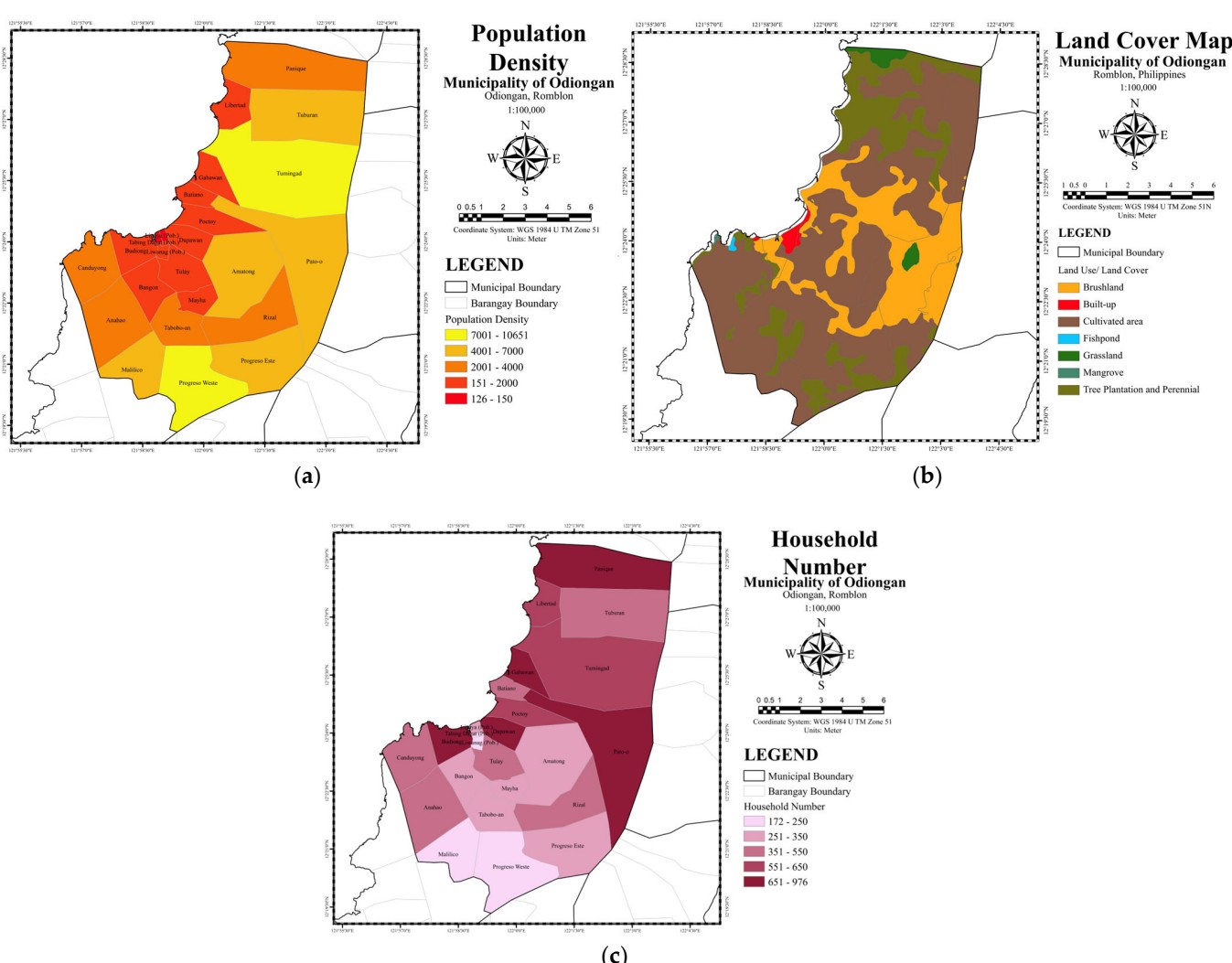

**Figure 8.** Result maps for flood exposure parameters in using ArcMap: (**a**) Population Density; (**b**) Land Use Map; and (**c**) Household Number.

Figure 8a shows the Population Density in the Municipality of Odiongan. The map demonstrates that population density is low in the nearby Poblacion area or where the center of the municipality is located. Barangay Ligaya and Liwayway as the lowest population density followed by Tabin-Dagat, Dapawan, Budiong, Liwanag, Bangon, Tulay,

Mayha, Batiano, Gabawan, and Libertad. However, Barangay Tumingad and Progresso Weste displayed a high population density.

Land use and land cover type are vital factors responsible for flood incidence. The land cover represents the physical (water, bare ground, and artificial structures) and biological (grass and trees) land cover. In contrast, land use describes how men utilized the land to improve their state of living [58,59]. The occurrence of flooding was inversely related to vegetation density. The study area's land cover classes were prepared from the municipal's CLUP. The land cover map was reassigned by categorizing the land-use types into seven (7) general categories. The map shown in Figure 8b indicates that most of the municipality's land area was cultivated. Brushland is observed in the center part of the map, while the scattered site of tree plantation and perennial were seen on the map. The built-up zone is located in the town proper of the municipality.

Evacuation decision before (preemptive) or during (forced) a disaster indicates the choice of households to evacuate or stay in the area at risk of impending hazard [43]. The household number was considered one of the parameters in assessing exposure. The study obtained the data from PSA's last 2020 census. Presented in Figure 8c is the range of the number of households for every barangay in the municipality. It was recorded that a high number of households in Baranagay, Panique, Pato-o, Gabawan, Dapawan, and Tabin-Dagat were exposed to flood events. Between 172 and 250 households, which was the least number, were observed in Liwanag, Liwayway, Barangay Malilico, and Progresso Weste. However, Liwanag and Liwayway have small land areas that cater only to some houses.

### 3.2. Evaluation and Assessment of Parameters

Contributing factors were evaluated and assessed.

The decision was segregated into its independent components. It was presented in a hierarchy diagram of at least three levels: goal, criteria, and indicators. The study structure using AHP was shown in Figure 9, wherein the uppermost place of the hierarchy is the primary goal of having a flood risk map. The lower level of the order contains the criteria contributing to attaining the goal: flood hazard map, flood vulnerability map, and flood exposure map. Finally, the lowest level included the indicators: average annual rainfall, elevation, slope, flood depth, soil type, gender ratio, individual age, average income, number of PWD, highest educational attainment, water usage, emergency preparedness, types of built-up structures, distance to evacuation area, population density, land cover, and number of households. The featured weight was assigned for each parameter, where were reclassified and normalized. Assigned values depend on the type of level or category. Table 2 indicates the feature weight of every indicator. The results of the weights computed using the AHP based on experts' inputs are shown in Table 3. These are the final weights of each parameters identified through AHP and was ensured to pass the consistency index requirement for it to be considered as valid.

**Table 2.** Standard matrix for hazard parameters.

|  | AAR | E | S | ST | FD | Weights | Percentage Weights |
|---|---|---|---|---|---|---|---|
| **AAR** | 0.224979 | 0.221826 | 0.259802 | 0.194982 | 0.228034 | 0.225925 | 23% |
| **E** | 0.208256 | 0.205338 | 0.197392 | 0.202930 | 0.208920 | 0.204567 | 20% |
| **S** | 0.154522 | 0.185622 | 0.178439 | 0.193329 | 0.188299 | 0.180042 | 18% |
| **ST** | 0.196749 | 0.172539 | 0.157384 | 0.170516 | 0.156328 | 0.170703 | 17% |
| **FD** | 0.215493 | 0.214675 | 0.206983 | 0.238243 | 0.21842 | 0.218763 | 22% |
|  |  |  |  |  |  |  | 100% |

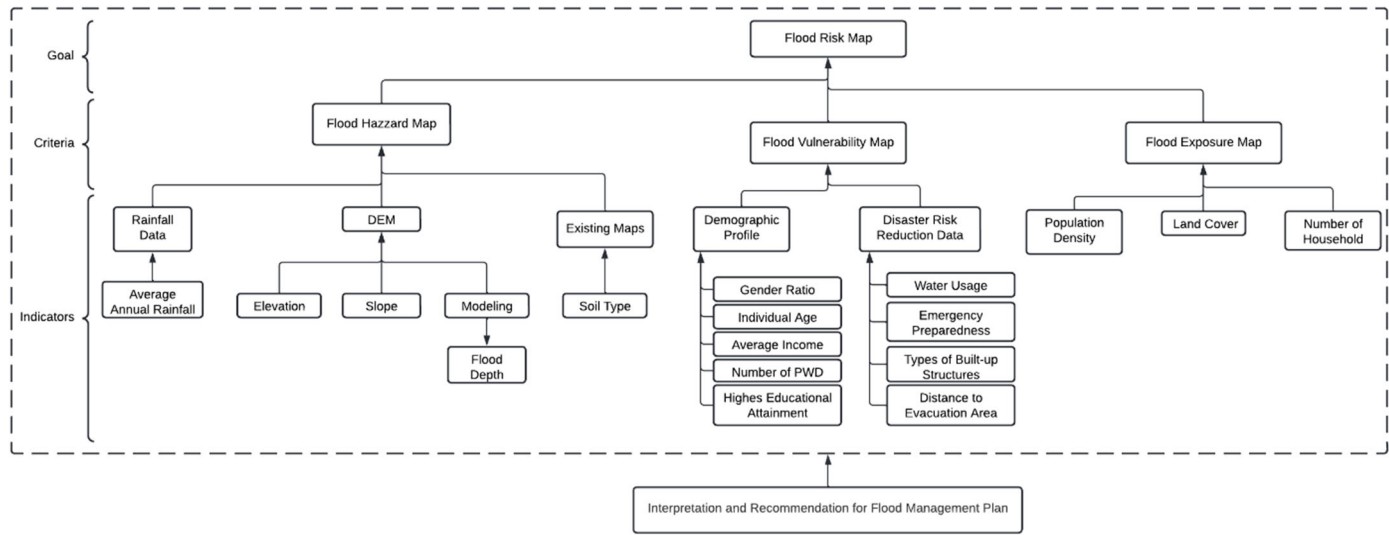

**Figure 9.** Multicriteria data analysis for flood risk assessment in AHP framework.

**Table 3.** Standard matrix vulnerability parameters.

|  | GR | MA | AI | NPWD | HEA | WU | EP | TBS | DEA | Weights | Percentage Weights |
|---|---|---|---|---|---|---|---|---|---|---|---|
| **GR** | 0.091812 | 0.072630 | 0.083258 | 0.095499 | 0.093304 | 0.088862 | 0.116943 | 0.095699 | 0.080989 | 0.091000 | 9% |
| **MA** | 0.136601 | 0.108062 | 0.134171 | 0.116910 | 0.076277 | 0.083485 | 0.110578 | 0.090066 | 0.125462 | 0.109068 | 11% |
| **AI** | 0.124171 | 0.090691 | 0.112602 | 0.090232 | 0.105573 | 0.129437 | 0.126444 | 0.135126 | 0.096882 | 0.112351 | 11% |
| **NPWD** | 0.101895 | 0.097966 | 0.132265 | 0.105988 | 0.102919 | 0.107756 | 0.086492 | 0.107974 | 0.106007 | 0.105474 | 11% |
| **HEA** | 0.078704 | 0.113313 | 0.085309 | 0.082368 | 0.079983 | 0.095524 | 0.073935 | 0.096588 | 0.092532 | 0.088695 | 9% |
| **WU** | 0.109121 | 0.136708 | 0.091879 | 0.103882 | 0.088433 | 0.105616 | 0.085829 | 0.111776 | 0.117633 | 0.105653 | 11% |
| **EP** | 0.108148 | 0.134616 | 0.122671 | 0.168801 | 0.149019 | 0.169508 | 0.137751 | 0.134197 | 0.130969 | 0.139520 | 14% |
| **TBS** | 0.114932 | 0.143735 | 0.099830 | 0.117594 | 0.099203 | 0.113196 | 0.122971 | 0.119798 | 0.130779 | 0.118004 | 12% |
| **DEA** | 0.134616 | 0.102279 | 0.138016 | 0.118726 | 0.205288 | 0.106616 | 0.139056 | 0.108777 | 0.118748 | 0.130236 | 13% |
|  |  |  |  |  |  |  |  |  |  |  | 100% |

The final relative weights of the alternatives which were defined by computing the product's linear combination (LC) between the relative weight of each criterion and the alternative for that specific criterion are shown in Tables 2–4, whereas Tables 5–7 are the computation of CI and CR for hazard, vulnerability, and exposure, respectively. A consistency ratio of 1.32%, 3.31%, and 6.85% were noticed in the hazard, vulnerability, and exposure below 10%. The experts made repeated responses to obtain the acceptable CR for all judgments. Further, final weights for hazard, vulnerability, and exposure are shown in Table 8. These weights are integrated into ArcGIS to generate hazard, vulnerability, and exposure maps with the corresponding index value.

**Table 4.** Standard matrix for exposure parameters.

|  | PD | LC | NH | Weights | Percentage Weights |
|---|---|---|---|---|---|
| **PD** | 0.326245 | 1/3 | 1/3 | 0.327369 | 33% |
| **LC** | 0.37159 | 0.34364 | 0.31783 | 0.344357 | 34% |
| **NH** | 0.302161 | 0.354651 | 0.328012 | 0.328274 | 33% |
|  |  |  |  |  | 100% |

**Table 5.** Computation of CR and CI of hazard parameters for consistency of AHP.

|       | AAR      | E        | S        | ST       | FD       | Sum      | Crit. Weigths |
|-------|----------|----------|----------|----------|----------|----------|---------------|
| AAR   | 0.225925 | 0.244066 | 0.328938 | 0.258341 | 0.235870 | 1.293139 | 5.72376382    |
| E     | 0.189362 | 0.204567 | 0.226295 | 0.243454 | 0.195670 | 1.059347 | 5.17847996    |
| S     | 0.123658 | 0.162756 | 0.180042 | 0.204129 | 0.155214 | 0.825800 | 4.58669759    |
| ST    | 0.149284 | 0.143437 | 0.150561 | 0.170703 | 0.122176 | 0.736161 | 4.31251769    |
| FD    | 0.209539 | 0.228710 | 0.253757 | 0.305653 | 0.218763 | 1.216422 | 5.56046026    |
|       |          |          |          |          |          | $y_{max}$ | 5.07238386   |
|       |          |          |          |          |          | CI       | 0.018095966   |
|       |          |          |          |          |          | CR       | 0.016157112   |

**Table 6.** Computation of CR and CI of vulnerability parameters for consistency of AHP.

|      | GR       | MA       | AI       | NPWD     | HEA      | WU       | EP       | TBS      | DEA      | Sum      | Crit. Weights |
|------|----------|----------|----------|----------|----------|----------|----------|----------|----------|----------|---------------|
| GR   | 0.091000 | 0.061162 | 0.067285 | 0.081994 | 0.106155 | 0.076564 | 0.077254 | 0.072694 | 0.062064 | 0.696171 | 7.650275      |
| MA   | 0.162276 | 0.109068 | 0.129960 | 0.120308 | 0.104014 | 0.086214 | 0.087554 | 0.081999 | 0.115235 | 0.996628 | 9.137668      |
| AI   | 0.151949 | 0.094290 | 0.112351 | 0.095649 | 0.148296 | 0.137692 | 0.103129 | 0.126726 | 0.091663 | 1.061744 | 9.450256      |
| NPWD | 0.117057 | 0.095620 | 0.123891 | 0.105474 | 0.135719 | 0.107612 | 0.066225 | 0.095063 | 0.094157 | 0.940818 | 8.919941      |
| HEA  | 0.076032 | 0.093005 | 0.067196 | 0.068929 | 0.088695 | 0.080220 | 0.047605 | 0.071511 | 0.069114 | 0.662308 | 7.467245      |
| WU   | 0.125572 | 0.133660 | 0.086208 | 0.103554 | 0.116815 | 0.105653 | 0.065829 | 0.098578 | 0.104661 | 0.940530 | 8.902074      |
| EP   | 0.164345 | 0.173804 | 0.151996 | 0.222206 | 0.259944 | 0.223923 | 0.139520 | 0.156289 | 0.153880 | 1.645906 | 11.79692      |
| TBS  | 0.147721 | 0.156959 | 0.104619 | 0.130927 | 0.146361 | 0.126474 | 0.105343 | 0.118004 | 0.129960 | 1.166368 | 9.884114      |
| DEA  | 0.190954 | 0.123266 | 0.159630 | 0.145889 | 0.334267 | 0.131470 | 0.131470 | 0.118254 | 0.130236 | 1.465436 | 11.25217      |
|      |          |          |          |          |          |          |          |          |          | $y_{max}$ | 9.384519     |
|      |          |          |          |          |          |          |          |          |          | CI       | 0.048065      |
|      |          |          |          |          |          |          |          |          |          | CR       | 0.033148      |

**Table 7.** Computation of CR and CI of exposure parameters for consistency of AHP.

|      | PD       | LC       | NH       | Sum      | Crit. Weights |
|------|----------|----------|----------|----------|---------------|
| PD   | 0.327369 | 0.287416 | 0.353462 | 0.968247 | 2.957665      |
| LC   | 0.392224 | 0.344357 | 0.33367  | 1.070251 | 3.10797       |
| NH   | 0.30404  | 0.338789 | 0.328274 | 0.971104 | 2.958207      |
|      |          |          | $y_{max}$ | 3.007947 |              |
|      |          |          | CI       | 0.003974 |               |
|      |          |          | CR       | 0.006851 |               |

**Table 8.** Final weights and percentage weights of every parameter for hazard, vulnerability, and exposure.

| Parameters                              | Weights  | Percentage Weights |
|-----------------------------------------|----------|--------------------|
| **Hazard Parameters**                   |          |                    |
| Average Annual Rainfall (AAR)           | *0.225925* | 22.59%           |
| Elevation (E)                           | *0.204567* | 20.46%           |
| Slope (S)                               | *0.180042* | 18.0%            |
| Soil Type (ST)                          | *0.170703* | 17.07%           |
| Flood Depth (FD)                        | *0.218763* | 21.88%           |
| **Vulnerability Parameters**            |          |                    |
| Gender Ratio (GR)                       | *0.091000* | 9.1%             |
| Mean Age (MA)                           | *0.109068* | 10.91%           |
| Average Income (AI)                     | *0.112351* | 11.24%           |
| Number of PWD (NPWD)                    | *0.105474* | 10.55%           |
| Highest Educational Attainment (HEA)    | *0.088695* | 8.87%            |

**Table 8.** *Cont.*

| Parameters | Weights | Percentage Weights |
|---|---|---|
| Water Usage (WU) | *0.105653* | 10.57% |
| Emergency Preparedness (EP) | *0.139520* | 13.95% |
| Types of Build-up Structures (TBS) | *0.118004* | 11.8% |
| Distance to the nearest Evacuation Area (DEA) | *0.130236* | 13.02% |
| **Exposure Parameters** | | |
| Population Density (PD) | *0.327369* | 32.74% |
| Land Use/Land Cover (LULC) | *0.344357* | 34.44% |
| Household Number (HN) | *0.328274* | 32.83% |

### 3.3. Development of Flood Risk Map

The visualization outputs for hazard, vulnerability, and exposure are shown in Figure 10. These maps were generated after computing the criteria weights using AHP and incorporating these weights with a GIS-based process consisting of overlays, raster conversion, and layer clipping.

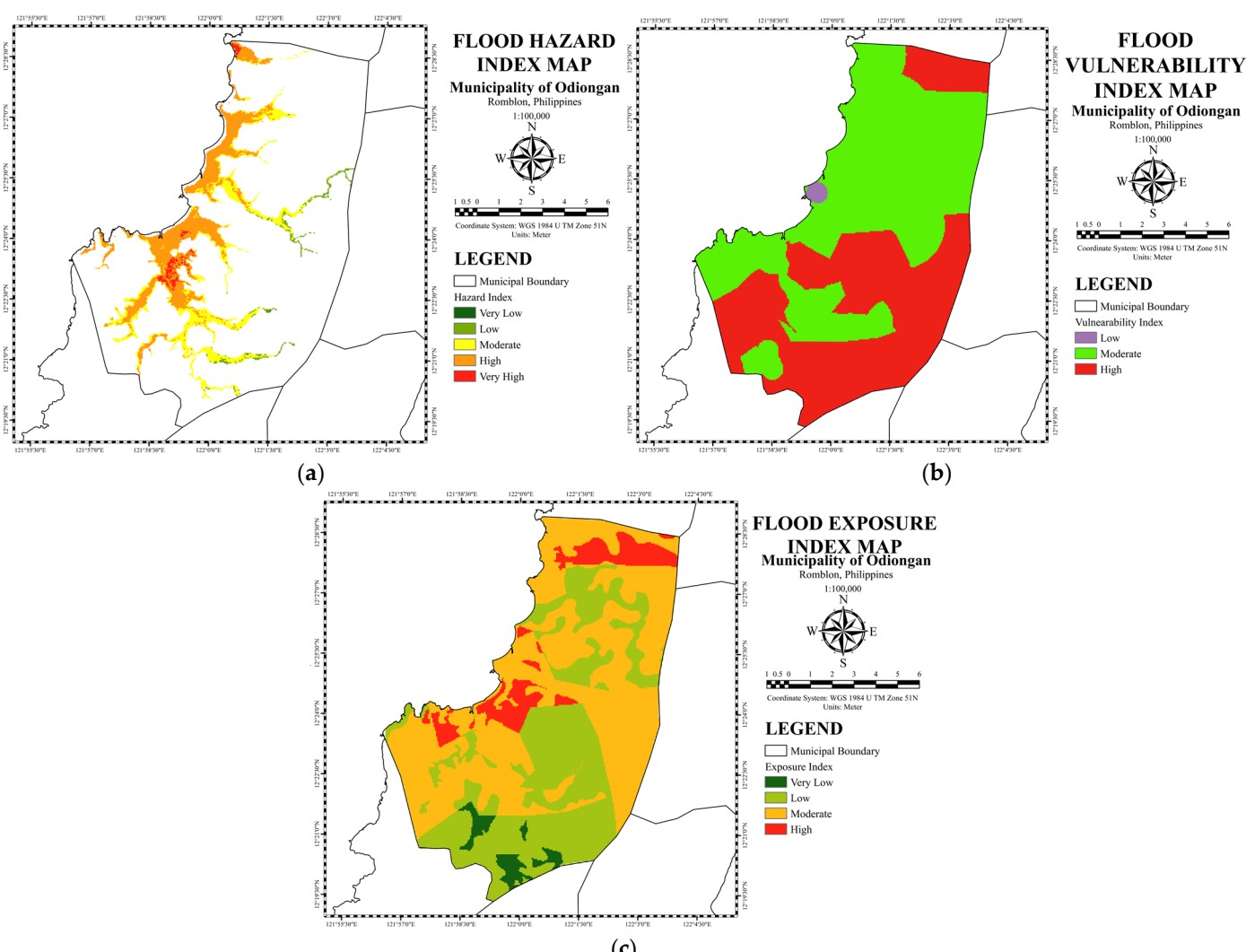

**Figure 10.** (**a**) Flood Hazard, (**b**) Flood Vulnerability, (**c**) Flood Exposure Index Maps of the Odiongan, Romblon, integrating all parameters in ArcGIS.

As shown in Figure 10a, the flood hazard index map, which combined all five (5) factors, was developed using the overlay tool in ArcGIS. The hazard map was classified into five (5) levels: very low (green), low (yellow-green), moderate (yellow), high (orange), and very high (red) and covers 0.009, 1.07, 8.74, 11.71, and 0.76 square kilometers, respectively. It was observed that areas with hazard index values were mostly affected according to the flood depth map, where river bodies are also located. Moderate to very high hazard is seen in areas of Poblacion, including Poctoy, Bangon, and Anahao. However, a moderate to high index was presented in some parts of Batiano, Gabawan, Libertad, and Paniques.

The vulnerability index map obtained by combining all nine (9) parameters highlights five areas (low, moderate, and high), as shown in Figure 10b, using the overlay tool. The flood vulnerability map generated from ArcGIS determines the degree of susceptibility of the flood-prone zone [29]. Purple connotes low vulnerability, yellow-green for moderate, and red for high vulnerability. Low, moderate, and high classes cover 0.56%, 54.52%, and 44.92% of Odiongan. As can be seen, only barangay Batiano is in low vulnerability and barangays Mayha, Tabobo-an, Canduyong, Malilico, Poctoy, Gabawan, Tumingad, Tuburan, Libertad, and a portion of Poctoy and Budiong is in moderate vulnerability. Tulay, Amatong, Pato-o, Rizal, Progresso Este, Progresso Weste, Anahao, Bangon, Tulay, and Panique is observed as highly vulnerable.

Exposure parameters (population density, number of households, and land cover) were overlaid in ArcGIS to develop a Flood Exposure Index Map. The resulting map derived four categories (very low, low, moderate, and high) in flood exposure using experts' weights, as shown in Figure 10c. The green symbolizes very low exposure, yellow-green for low, orange for moderate, and red for high exposure, which covers 2.61%, 36.39, 53%, and 8.01%, respectively, of the total land area of Odiongan. The Poblacion area and parts of Budiong, Gabawan, Batiano, and Panique were at high exposure to flood. However, more than half of the map is scattered orange illustrating moderate exposure.

The result of the flood risk assessment is laid into a map for a better comprehension. As shown in Figure 11, the analysis result was a map combining flood hazard, flood vulnerability, and flood exposure index maps utilizing ArcGIS. Equal weights were employed in three (3) maps. The flood risk map is categorized using equal intervals with four (4) levels (very low, low, moderate, and high). In total, 93.92 square kilometers (green) are classified as very low risk, comprising 83.78% of the total land area. The yellow-green color as low risk covers approximately 0.198 square kilometers (0.15% of land area) and is seen in a small part of Barangay Rizal and Tumingad. Overall, 12.86% of the total area was categorized as moderate risk (yellow) and noted on the map as 17.56 square kilometers. A portion of barangays Rizal, Progresso Este, Progresso Weste, Malilico, Amatong, Anahao, Canduyong, Pato-o, Tumingad, Mayha, Tuburan, and Panique were observe with moderate risk. Gabawan, Batiano, Tabobo-an, and Libertad were also at moderate risk, with more than half of their respective barangay boundaries. The Poblacion area, Tulay, Bangon, Anahao, Dapawan, and Poctoy were at high risk to flood occurrence, covering 3.26% of land area (4.46 square kilometers). Parcels with high risk were also sighted in Canduyong, Gabawan, and Panique. Through this flood risk map, the municipal councils, planning agencies, and other stakeholders can prepare Flood Management Plan to reduce the threat to lives due to flooding and anticipate future infrastructure development in the municipality.

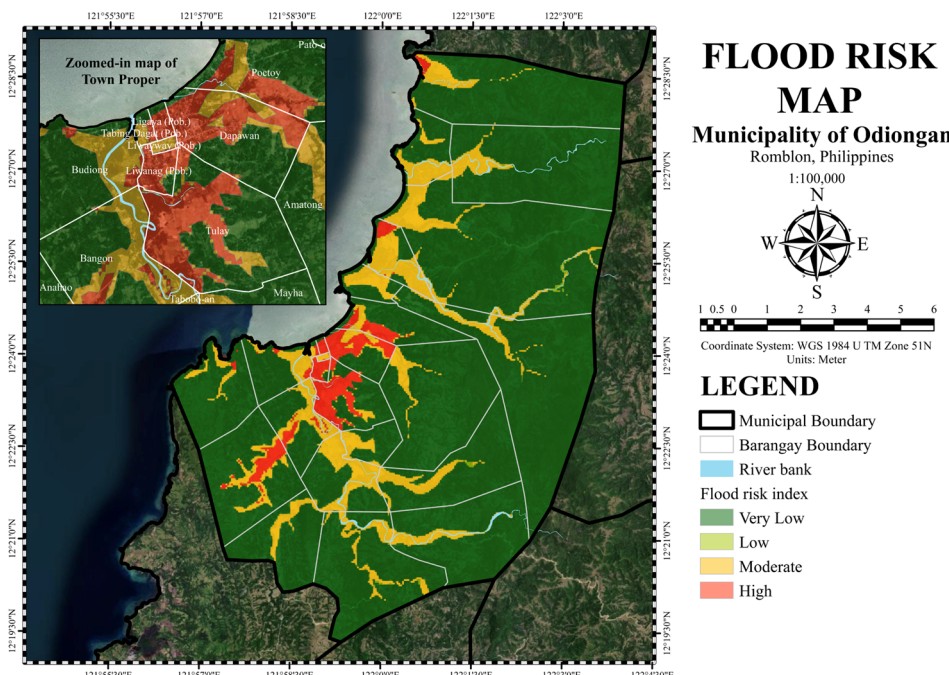

**Figure 11.** Imagery and flood risk map (50% transparency) with 1:50,000 scale zoomed-in map of town proper of the Municipality of Odiongan.

## 4. Discussion

The harmful effects of disasters in flood-prone areas have amplified in severity over the years. The damage ensuing from these events are also exponentially growing, where serious land use and climate change impacts are alarming. The development of a flood risk map utilizing elevation models, demographics, awareness, and disaster-risk-related data integrated into GIS and analyzed using AHP is an effective tool for evaluating risk. Integrating available data through the archival, automated process, and survey data were possible in deriving each criterion. It is noted in the flood risk map that approximately 83.78% of the total area map is at very low risk, 0.15% is at low, 12.86% at moderate, and 3.26% for high flood risk. 93.92, 0.198, 17.56, and 4.46 square kilometers are very low, low, moderate, and high risk to flood. A small portion of Barangay Rizal and Tumingad are at low risk (yellow-green). Moderate risk was noted on the map covering a part of barangays Rizal, Progresso Este, Progresso Weste, Malilico, Amatong, Anahao, Canduyong, Pato-o, Tumingad, Mayha, Tuburan, and Panique. Then, Gabawan, Batiano, Tabobo-an, and Libertad were at moderate risk covering half of their respective barangay boundaries. Moreover, based on the zoomed-in map of the town proper, the Poblacion area, Tulay, Bangon, Anahao, Dapawan, and Poctoy are at high risk to flood occurrence. The possible reason for this is the rapid urbanization and infrastructure development in the town proper of Odiongan, as shown in Figure 11. In addition, moderate to high-risk indexes were observed along the riverbanks of Odiongan. In validation, areas at high risk are known to have flood events. The results of the risk maps urge the municipality to plan for flood mitigation or develop a comprehensive flood management plan as a countermeasure during for future flood events. Flood risk assessment is required for the flood management and mitigation of cities and municipalities. However, the specific parameters available in such areas are significantly different, which differs the risk assessment index weight. The study of Cai et al. [15] adopted the frequency of rainstorms and average annual precipitation of counties as hazard parameters, similar to what were used in this study. Average annual rainfall represented 22.59% of the weight of the total hazard parameters which showed its importance or role in identifying hazard level in a certain area. For vulnerability factors, the study considered the following: population density, average area GDP, per capita disposable income, road network density, and land use type. Average income was

basically identified to have the highest weight among all other factors considered in the vulnerability index with 11.24%, this just shows that the capability to spend affects how people could prepare and protect themselves during a disaster. A risk assessment could also be performed, even in mega-cities, as based on the study of Lyu et al. [60]. Parameters such as rainy season, average rainfall, average rainy day for hazard and elevation, slope, river proximity, river density as exposure parameters; and land use, metro line proximity, metro line density, road network proximity and road network density as vulnerability indicators were considered, similarly to what was considered in this study. This shows that methodology used in this study could also be applied in mega-cities here in the Philippines with some revision of factors to be considered which will be based on the city's characteristics. A local study [46] used Confirmatory Factor Analysis (CFA) to simultaneously investigate the interrelationship between vulnerability to natural hazards composed of exposure, sensitivity, and resilience. Additionally, a study [61] developed a comprehensive framework for vulnerability assessment to determine vulnerability that would deliver a transparent understanding and improve community competency leading to the development of methodologies to assess factors and indicators of vulnerability. For this assessment, the combination of hazard, vulnerability, and exposure assessment established a clear definition of risk levels [36,62]. Compared to other related studies, the evaluation used the Sendai Framework to clear out the true meaning of disaster risk based on all dimensions of vulnerability, exposure, and hazard characteristics of the environment. Regarding hazard assessment, many studies used factors such as topographical [63], natural, and anthropogenic [62] factors. In this study, factors used were average annual rainfall, slope, elevation, soil type, and simulated flood depth, which were decided according to the data availability. In terms of vulnerability assessment, flood vulnerability is affected by factors such as settlement conditions, infrastructure, policy and capacities of the authorities, social inequities, and economic patterns. The study effectively generated a vulnerability map integrating age, gender ratio, average income, individual physical health, educational attainment, emergency preparedness, and types of built-up structures as factors. Exposure assessment of the study is based on the analysis of [64], which used land use and population density as parameters. The study added the additional factor, household number, to justify the flood exposure.

The accuracy of generating flood hazard maps is highly dependent on the quality of topographical data [62]. Topographical information such as DEMs is an excellent source to derive topographic factors responsible for flood activity [65]. Although local studies [63,64] utilized LiDAR-derived DEM due to its inherent high vertical accuracy and resolution, If-SAR DTM from NAMRIA, as one of the highest resolution available DEM in the location, is used in the study, which showed reasonably consistent with the generated maps.

The flood risk assessment map demonstrates flood risk areas that must be managed on a priority basis. In some studies, different methodologies were established for assessing flood risk. One uses a hybrid intelligence model [14], probability [66,67], polygon approach [68], Quantitative risk assessment methods (e.g., Fine Kiney [55] and Maxent model [8]), and MCDA techniques, such as fuzzy majority approach [30], fuzzy variable set theory [44], multi-attribute value, frequency ratio, artificial neural network [65], fuzzy analytical hierarchy process (FAHP) [41], and decision tree [69] to assess the flood risk. Most government units, municipal planners, and other concerned agencies use AHP as an MCA in terms of disaster risk reduction, land use plans, and decisions requiring a comprehensive judgment and recommendation from experts to benefit the community. Using the AHP decision-making method for the multiple flood-related factors is extensively adopted. From the result of the study, it is noticed that AHP proposes a flexible, stepwise, and precise process of analyzing complicated problems in an MCDA environment. In addition, the resolution of complications in multicriteria methods is realized as the primary use of AHP. In this research, where three (3) criteria (hazard, vulnerability, and exposure) with multiple parameters (5, 9, 3, respectively) have different dimensions, it makes a simple MCDA dilemma more complicated. In this study, the AHP-based flood risk assessment

method is considered relatively practical, convenient, and promotes interactive usage by flood managers for continuing improvement. AHP is a useful method for selecting contending options in light of a range of objectives to be convened. The computations were not complex, and the MCA did not need to understand the calculation to use the procedure. Nonetheless, the AHP has established considerations because it highlights the knowledge of decision-makers' preferences [32].

Compared to other flood maps available online and offline, the study highlighted how comprehensive the methodology is. Government agencies' flood maps consider only one to two criteria (hazard, susceptibility, or only topographical related data) that limit the assessment's accuracy. For this research, several social, economic, environmental, and technical factors were considered to develop a comprehensive flood risk map. Future studies could be performed to improve this risk map where environmental quality could be incorporated [70], such as flooding, was found to be correlated with the increasing concentration of manganese in Marinduque. Considering environmental quality in risk assessment would be useful.

## 5. Conclusions

The number of flooding events in the Municipality of Odiongan caused property damage to the community and has put lives at risk based on historical documents. This showed how vital flooding risk assessments using GIS-based, and multi-criteria decisions are. The results of this study are useful for improving the municipality's flood mitigation and risk management strategies.

This study assessed the flood risk in the Municipality of Odiongan, Romblon, considering relevant factors in floods using AHP and following the Sendai Framework. The study used ArcGIS, HEC-RAS, and HEC-HMS to map and model the primary and secondary data as parameters that mainly contribute to flooding. The study considered the following parameters: average annual rainfall, elevation, slope, soil type, and flood depth for hazard criteria; gender ratio, mean age, average income, PWD, educational attainment, water usage, emergency preparedness, type of built-up structures, and distance to evacuation area in vulnerability and population density, land cover and household number for exposure, respectively. Each parameter was compared to one another by pairwise comparison to identify its weights based on experts' judgment and integrate these weights of factors into AHP. Weights were computed as follows: average annual rainfall with 23%, elevation—20%, slope—18%, soil type—17%, and flood depth—22% for hazard criteria; gender ratio—9%, mean age—11%, average income—11%, PWD—11%, educational attainment—9%, water usage—11%, emergency preparedness—14%, type of built-up structures—12%, and distance to evacuation area—13% in vulnerability and population density with 33%, land use—34% and household number—33% for exposure. It was noted that approximately 83.78% of the total area map was at very low risk, 0.15% is at low, 12.86% at moderate, and 3.26% for high flood risk. Then, 93.92, 0.198, 17.56, and 4.46 square kilometers were very low, low, moderate, and high risk to flood. The risk assessment results derived a flood risk map which found out the nine (9) barangays were at high risk of flooding, notably the Poblacion Area, Tulay, Bangon, Tabobo-an, Dapawan, and Anahao. The flood risk map developed in this study considered the social, economic, environmental, and technical factors that represent those factors in actual scenarios. Highlighting its difference compared to the flood depth map or available flood maps online and released by the different concerned agencies incorporating the topographic aspect of the area.

In conclusion, the result of this flood risk assessment is essential for the municipality to improve their flood management strategies considering the risk factors: hazard, vulnerability, and exposure. It can help the planning agencies and other stakeholders anticipate flood risk, especially the LGU, by integrating the output into their CLUP. This technique can also be employed by other local government units to come up with more practical and effective strategies. Moreover, future studies must be conducted to enhance and update the flood risk assessment methods and management.

**Author Contributions:** Conceptualization, J.G.G. and C.E.F.M.; methodology, J.G.G. and C.E.F.M.; software, J.G.G.; validation, J.G.G.; formal analysis, J.G.G., C.E.F.M. and D.B.S.; investigation, J.G.G.; resources, J.G.G. and C.E.F.M.; data curation, J.G.G. and C.E.F.M.; writing—original draft preparation, J.G.G.; writing—review and editing, J.G.G., C.E.F.M., D.B.S. and F.J.T.; visualization, J.G.G. and C.E.F.M.; supervision, C.E.F.M., D.B.S. and F.J.T.; project administration, J.G.G. and C.E.F.M. All authors have read and agreed to the published version of the manuscript.

**Funding:** The study was funded by the Department of Science and Technology—Engineering Research and Development for Technology and Mapúa University.

**Informed Consent Statement:** Informed consent was obtained from all subjects involved in the study.

**Data Availability Statement:** All data are contained in the manuscript.

**Acknowledgments:** The author would like to acknowledge the in-kind support of Romblon State University, Local Government of Odiongan, National Mapping and Resource Information Authority (NAMRIA), Philippine Atmospheric, Geophysical, and Astronomical Services Administration (PA-GASA), Mines and Geosciences Bureau (MGB), Bureau of Soils and Water Management (BSWM), and the Philippine Statistics Authority (PSA) for the conduct of the study.

**Conflicts of Interest:** The authors declare no conflict of interest.

**Appendix A**

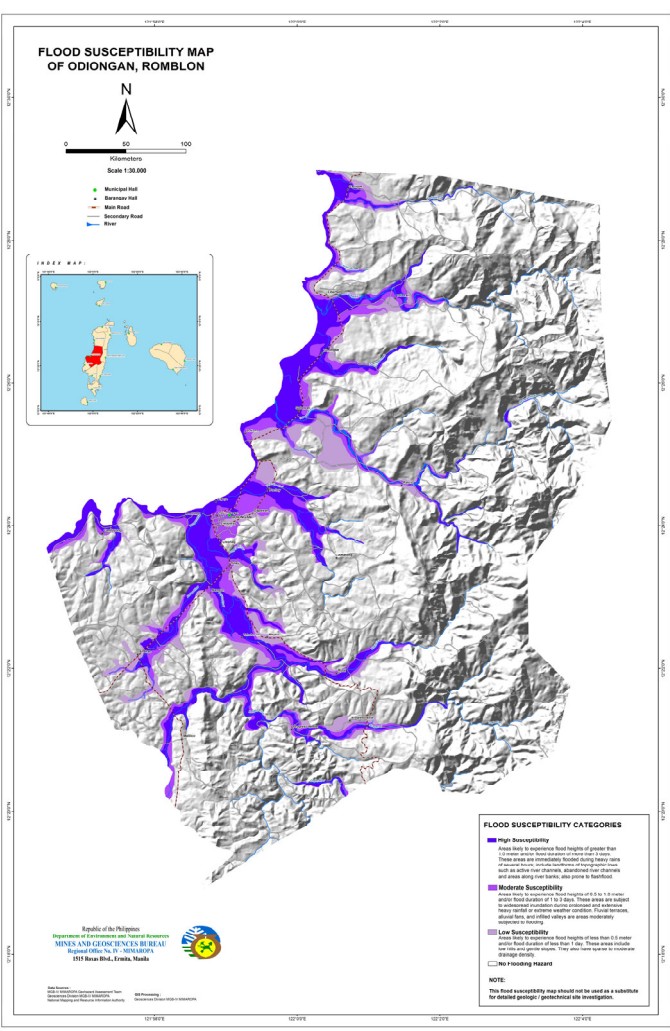

**Figure A1.** Flood Susceptibility Map from MGB-DENR.

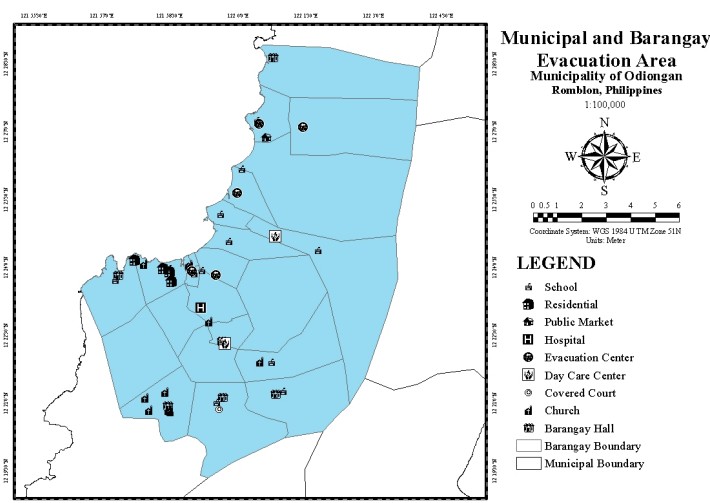

**Figure A2.** Identified evacuation area of barangay in Odiongan, Romblon, during emergency events.

In identifying the weights of factors, the pairwise comparison procedure usually contains a questionnaire for comparing all the elements and a geometric mean to arrive at a final solution. Psychologists conclude that the nine points shown in Table A1 are the most used comparison matrix individuals can compare simultaneously and consistently rank.

**Table A1.** The nine-point intensity of importance scale was modified from Schoenherr. Copyright 2008 Elsevier.

| Intensity of Importance | Definition | Description |
| --- | --- | --- |
| 1 | Equally important | Two factors contribute equally to the objective |
| 3 | Moderately more important | Experience and judgment slightly favor one over the other |
| 5 | Strongly more important | Experience and judgment strongly favor one over the other |
| 7 | Very strong, more important | Experience and judgment very strongly favor one over the other. Its importance is demonstrated in practice. |
| 9 | Extremely more important | The evidence favoring one over the other is of the highest possible validity. |
| 2, 4, 6, 8 | Intermediate values | When compromise is needed. |
| Reciprocals of above | If an element *i* has one of the above numbers assigned to it when compared with element *j*, then *j* has the reciprocal value when compared with *i* | |
| Ratios (1.1–1.9) | If the activities (elements) are very close. | It may be challenging to assign the best value, but when compared with other contrasting activities (elements), the size of the small numbers would not be too noticeable, yet they can still indicate the relative importance of the activities (elements) |

The feature weight was assigned to each parameter, where levels were reclassified and normalized into 1, 2, 3, 4, and 5 (1 for the least priority and 5 for the most prior). Assigned values depend on how primary the level or category is. Table A2 indicates the feature weight of every indicator.

**Table A2.** Parameters with their designated feature weight.

| Indicators | Feature Class | Feature Weight |
|---|---|---|
| **Flood Hazard Parameters** | | |
| Average Annual Rainfall (in mm) | 2200 | 1 |
| | 2210 | 1 |
| | 2220 | 2 |
| | 2230 | 3 |
| | 2240 | 4 |
| | 2250 | 5 |
| Elevation (in meters) | 0–5 | 5 |
| | 6–20 | 4 |
| | 21–50 | 3 |
| | 51–150 | 1 |
| | 151–600 | 0 |
| Slope (in degrees) | 0–3 | 5 |
| | 3–8 | 4 |
| | 8–18 | 3 |
| | 18–30 | 2 |
| | 30–50 | 1 |
| | 50 above | 0 |
| Soil Type | Sandy, loamy sand, or sandy loam | 1 |
| | Silt loam or loam | 3 |
| | Clay loam, silty clay loam, sandy clay, or clay | 5 |
| Flood Depth (in meters) | 0–0.5 | 1 |
| | 0.51–1 | 2 |
| | 1.01–1.5 | 3 |
| | 1.51–2 | 4 |
| | 2> | 5 |
| **Flood Vulnerability Parameters** | | |
| Gender Ratio (men to women ratio) | 0.839339–0.839655 | 1 |
| | 0.839656–0.963855 | 2 |
| | 0.963856–1.008065 | 3 |
| | 1.008066–1.040521 | 4 |
| | 1.040522–1.208661 | 5 |
| Mean Age | 29–30 | 1 |
| | 31–32 | 2 |
| | 33–34 | 3 |
| | 35–36 | 4 |
| | 37–38 | 5 |
| Average Income | 500,000 and over | 1 |
| | 250,000 to 499,999 | 1 |
| | 100,000 to 249,999 | 2 |
| | 60,000 to 99,999 | 3 |
| | 40,000 to 59,999 | 4 |
| | Less than 40,000 | 5 |
| Number of PWD | 5–12 | 1 |
| | 13–26 | 2 |
| | 27–37 | 3 |
| | 38–55 | 4 |
| | 56–70 | 5 |
| Highest Educational Attainment | College Graduate | 3 |
| | High School Graduate | 5 |

**Table A2.** *Cont.*

| Indicators | Feature Class | Feature Weight |
|---|---|---|
| Water Usage | Ground | 4 |
| | Piped | 5 |
| Emergency Preparedness | Prepared | 5 |
| | Well prepared | 4 |
| | Very well prepared | 3 |
| Types of Build-up Structures | Permanent | 3 |
| | Semi-permanent | 4 |
| | Temporary | 5 |
| Distance to the nearest Evacuation Area (in meters) | 2000 above | 5 |
| | 2000 | 4 |
| | 1500 | 3 |
| | 1000 | 2 |
| | 500 | 1 |
| **Flood Exposure Parameters** | | |
| Population Density | 7001–10,651 | 1 |
| | 4001–7000 | 2 |
| | 2001–4000 | 3 |
| | 151–2000 | 4 |
| | 126–150 | 5 |
| Land Use and Land Cover | Brushland | 1 |
| | Built-up | 4 |
| | Cultivated Area | 3 |
| | Fishpond | 5 |
| | Grassland | 2 |
| | Mangrove | 0 |
| | Tree Plantation and Perennial | 0 |
| Household Number | 172–250 | 1 |
| | 251–350 | 2 |
| | 351–550 | 3 |
| | 551–650 | 4 |
| | 651–976 | 5 |

Pairwise comparison was based on adequate information, expert knowledge, and experience using a questionnaire. Ten (10) experts on-field and end-users determined the relevance of one alternative over the other with a pairwise comparison method presented in a matrix. Gathered weight for each criterion and option used a pairwise comparison technique. Then, each comparison is graded by expert respondents and end-user using the nine-point scale of importance. Eligible respondents and their credentials are shown in Table A3.

**Table A3.** Respondent's credentials for pairwise comparison technique.

| Respondent | Field of Expertise/Project Involvement | Agency/Institution/Project | Years in Service |
|---|---|---|---|
| 1 | Water Resource Engineering/Disaster Risk | Mapua University | 10 |
| 2 | Meteorology/Hydrology | PAGASA-DOST | 30 |
| 3 | Project Staff | FRAMER—Mapua University | 4 |
| 4 | Researcher | FRA Project—Asian Institute of Technology | 3 |
| 5 | Disaster Risk/Municipal Engineer | LGU—Odiongan | 30 |

**Table A3.** *Cont.*

| Respondent | Field of Expertise/Project Involvement | Agency/Institution/Project | Years in Service |
|---|---|---|---|
| 6 | Meteorology/Hydrology | Visayas State University—Department of Meteorology | 8 |
| 7 | Meteorology/Hydrology | Central Luzon State University | 3 |
| 8 | Senior Research Specialist | UP Training Center for Applied Geodesy and Photogrammetry | 5 |
| 9 | Agriculturist II/Regional Head | Department of Agriculture—Bureau of Soil and Water Management | 3 |
| 10 | Supervising Geologist/Expert in Landslide and flood susceptibility mapping | Department of Environment and Natural Resources—Mines and Geosciences Bureau MIMAROPA | 15 |

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
