# Peer review of "Flood Risk Assessment Using GIS-Based Analytical Hierarchy Process in the Municipality of Odiongan, Romblon, Philippines"

_applsci, doi:10.3390/app12199456_

Round 1

Reviewer 1 Report

Before it goes for acceptance, the following clarifications is needed:

1- The latest date of the data collection of the flood risk assessment? Reference?

2- Number of comparisons equation where it is used in the analysis?

3- Identification of factors should not be presented in the results section, should be shifted to the methods section.

4- Discussion part need to be more explained it is very short comparted to the results obtained.

Reviewer 2 Report

This manuscript treats about a flood risk assessment using GIS-based analytical hierarchy process in the municipality of Odiongan, Romblon, Philippines. The manuscript is interesting, novelty and well-written. I recommend to accept it subjected to the following modifications: 

- Lines 27-33. It is not necessary to indicate the weights in the abstract. The abstract must be brief and concise. 

- MCA, also called MCDM methods must be more explained in the introduction. For instance, mention that many MCA methods exists and why you have chosen AHP.  

- Line 106, change “[18] [19]” to “[18, 19]”. 

- Line 106, change “Suguruman [20]” to “Suguruman et al. [20]”. 

- Table 9. The quantity 20.46% is misaligned. 

- Review the format of the references according to the rules of the journal. 

- Appendix A. Please, include a text explaining something about the tables instead listing the tables without explicative text.

Reviewer 3 Report

STRUCTURE

-        English writing needs to be significantly improved. Readers may have troubles in reading it.

-        All the sections were redundantly long. Many of them can be omitted if the authors spend time to elaborate the content.

-        There are too many insignificant figures and tables as well.

CONTENT

-        Abstract was scientifically inaccurate. It just presented general introduction to the case study. It should discuss the significance of the performed research. Conclusions need to be referred to as well. You do not need to discuss results here (it is not helpful to present percentages again and again in the abstract). Instead, you can discuss how the results are useful to other scholars and/or practitioners.

-        Section 1 attempted to present introduction to the case study and literature review at the same time. As a result, nether of them was well explained. The case study should be introduced in Section 2. Literature review was weak and ambiguous. It was too long and too few references. It should be concise and accurate with a sizable number of recent references.

-        Discussion and conclusions were too long and provided too little interesting information.

-        In my opinion, it might be of potential but first, it needs to be fully restructured and rewritten. It was very hard for me to follow the authors’ logics due to inaccurate English writing and the random unorganised structure of the manuscript.

Round 2

Reviewer 3 Report

Authors must highlight the changes that they made from the previous manuscript. Otherwise, it is impossible for reviewers to continue review. If they meant that the entire manuscript has been renewed, it should be regarded as a new submission. 

Round 3

Reviewer 3 Report

My comments have been addressed. There hasnt been much improvement  from the previous manuscript.